# Impact of Turbulence on Aeolian Particle Entrainment: Results from Wind-tunnel Experiment

Jie Zhang[1,2*], Guang Li[3,4*], Li Shi[1,2], Ning Huang[1,2], and Yaping Shao[5]

[1]Key Laboratory of Mechanics on Disaster and Environment in Western China, Lanzhou University, Lanzhou 730000, China
[2]College of Civil Engineering and Mechanics, Lanzhou University, Lanzhou 730000, China
[3]College of Atmospheric Science, Lanzhou University, Lanzhou 730000, China
[4]College of Architecture Civil and Environmental Engineering, Ecole Polytechnique Federal de Lausanne, Lausanne 1015, Switzerland
[5]Institute for Geophysics and Meteorology, University of Cologne, Cologne 50923, Germany
[*]These authors contributed equally to this work.

**Correspondence:** Yaping Shao (yshao@uni-koeln.de)

**Abstract.** We hypothesize that large eddies play a major role in the entrainment of aeolian particles. To test this, wind-tunnel experiments are carried out to measure the particle entrainment rate for various sizes and flow conditions. Wind tunnel flows are usually neutrally stratified with no large eddies which typically seen in convective atmospheric boundary layers. Here, a novel technique is applied, by deploying a piece of randomly fluttering cloth, to generate large eddies similar to convective eddies, here referred to as quasi-convective turbulence. The characteristics of quasi-convective turbulence are analyzed with respect to neutral turbulence in the Monin-Obukhov similarity framework, and the probability distributions of surface shear stress are examined. We show that for given mean flow speed and in comparison with neutral flow conditions, quasi-convective turbulence increases the surface shear stress and alters its probability distribution, and hence substantially enhances the entrainment of sand and dust particles. Our hypothesis is thus confirmed by the wind-tunnel experiments. We also explain why large eddies are important to aeolian entrainment and transport.

## 1 Introduction

The entrainment of sand and dust particles is among the most important quantities to determine in aeolian studies. Based on Bagnold (1941), Owen (1964) showed that the vertically integrated saltation flux $Q$, can be expressed as

$$Q = \begin{cases} c_0 \frac{\rho}{g} u_*^3 (1 - \frac{u_{*t}^2}{u_*^2}) & \text{for } u_* > u_{*t} \\ 0 & \text{otherwise} \end{cases} \tag{1}$$

where $c_0$ is the Owen coefficient, $\rho$ air density, $g$ acceleration due to gravity, $u_*$ friction velocity and $u_{*t}$ threshold friction velocity. By definition, $u_* = \sqrt{\tau/\rho}$ is a descriptor of the surface shear stress $\tau$. Althrough challenged by some researchers (e.g., Martin and Kok, 2017), it has been shown in numerous studies that Eq. (1) is valid in general, but $c_0$ scatters over a wide range if Eq. (1) is fitted to $Q$ observations (Gillette et al., 1996, 1997; Leys, 1998; Liu et al., 2018). While the large scatter of $c_0$ is not yet fully explained, it is most likely because both $u_*$ and $u_{*t}$ are stochastic variables and hence saltation driven by

atmospheric boundary-layer (ABL) flows are also turbulent (Butterfield, 1991, 1998). In this sense, Eq. (1) is valid only for mean quantities, $\overline{Q}$ and $\overline{u}_*$, if we assume $u_{*t}$ is constant [this assumption is sufficient for the purpose of our study, but see Shao (2008), Raffaele et al. (2016), Liu et al. (2018), and Li et al. (2020) for discussions on $u_{*t}$ as a stochastic variable].

Our question here is how turbulence influences aeolian processes, especially, the entrainment of sand and dust particles into the atmosphere. Numerous studies on aerodynamic sand and dust entrainment have been carried out (e.g., Greeley and Iversen, 1987; Anderson and Haff, 1988, 1991; Loosmore and Hunt, 2000; Doorschot and Lehning, 2002; Jia and Wang, 2021), but in most of these studies, flow is assumed to be steady and the effect of turbulence on the entrainment is not accounted for. In reality, the flows and flow-driven aeolian processes are both impossible to stabilize. Althrough people can analyze the relevant average variables, the existence of nonlinear relations (e.g., Eq. 1) makes it difficult to determine the quantitative relations between the average variables. Butterfield (1991, 1998) investigated the behavior of saltating grains in unsteady flows and found that both the frequency and strength of wind gusts influence the rate of sand transport. Stout and Zobeck (1997) observed that saltation intermittently occurs even when $u_* < u_{*t}$, a phenomenon known as saltation intermittency. Xuan (2004) reported that turbulence decreases $u_{*t}$ or the threshold wind velocity $u_t$ and increases saltation intensity and the dust entrainment rate. Klose and Shao (2012) and Klose et al. (2014) developed a parameterization scheme for dust emission by convective turbulence and explained how dust emission can be produced by large eddies in weak mean wind conditions. Based on extensive field measurements, Comola et al. (2019) showed that neglecting saltation intermittency causes biases in the model estimated saltation fluxes. Shao et al. (2020) reported that saltation in unstable ABLs is generally more fully developed than in stable ABLs.

Saltation intermittency in the sense of Stout and Zobeck (1997) is a special case of saltation fluctuation at $\overline{u}_* \sim u_{*t}$ with $\overline{u}_*$ being the time averaged friction velocity. In general, if $u_* = \overline{u}_* + u_*^{'}$ (where $u_*^{'}$ is the $u_*$ derivation from $\overline{u}_*$) and $u_*^{'} > 0$, then we have $Q = Q(\overline{u}_*) + Q^{'}$ with $Q^{'} > 0$. The saltation intermittency Stout and Zobeck (1997) studied is for the case $\overline{u}_* = u_{*t}$ and $Q(\overline{u}_*) = 0$ but $Q^{'} > 0$. The above discussion suggests that the turbulent (or probabilistic) behavior of $u_*$ is of great importance to $Q$ and naturally also to sand and dust entrainment. Because the turbulent behavior of $u_*$ is closely related to ABL turbulence, our hypothesis is thus that ABL turbulence significantly influences the entrainment of sand and dust particles.

In stable and neutral ABLs, turbulence is generally weak and more homogeneous and isotropic, but in unstable (or convective) ABLs, turbulence is generally strong due to buoyancy production of turbulent kinetic energy and less homogeneous and isotropic due to the structure of large eddies. It is thus particularly interesting to study the influences of convective turbulence on aeolian processes. Khalfallah et al. (2020) pointed out that dust particle size at emission is dependent on ABL stability. Shao et al. (2020) compared some features of saltation and dust emission in convective and stable ABLs based on the field observations of the Japan-Australian Dust Experiment (JADE, Ishizuka et al., 2008, 2014), but due to the limitations of the field measurements, the role of turbulence in aerodynamic sand and dust entrainment could not be explicitly examined. We are thus motivated to acquire additional data to test our hypothesis.

Wind tunnel is a powerful tool for studying aeolian problems under controlled flow conditions (e.g., Rasmussen and Mikkelsen, 1991; Alfaro et al., 1997; Brown et al., 2008; Zhang et al., 2014). Although several methods have been proposed to generated turbulence in wind tunnels, including spires, roughness element, grid etc., all these methods are designed to increase

the intensity of turbulence in neutral ABLs, but are inadequate for generating large eddies similar to those commonly seen in convective ABLs. To generate convective turbulence inn wind tunnels usually requires the use of additional thermal forcing from the surface (e.g., EnFlo stratified flow wind tunnel, Hancock et al., 2013; Hancock and Farr, 2014; Hancock and Zhang, 2015; Hancock and Hayden, 2018),temperature control of recirculating air and floor panels (e.g., Inagaki et al., 2012; Zhang et al., 2013; Kanda and Yamao, 2016), thermally stratified wind tunnels (e.g., Marucci et al., 2018; Marucci and Carpentieri, 2020), etc. To apply surface heating requires normally a very large wind tunnel. To the best of our knowledge, studying the effect of convective eddies on aeolian processes has never been done in wind-tunnel experiments, mainly because we have so far no adequate means to generate convective turbulence in a wind tunnel for aeolian experiments. Here we apply a simple forced-perturbation technique (using a piece of randomly fluttering cloth) to generate quasi-convective turbulence, namely, turbulence in a neutrally stratified flow but with characteristics of convective turbulence. Aerodynamic sand and dust entrainment rates under various mean-wind and quasi-convective turbulence conditions are measured. We use these wind-tunnel data to study how and why turbulence influences the aerodynamic sand and dust entrainment rate.

## 2 Wind-tunnel Experiment and Instrumentation

We carried out the experiment in the Lanzhou University wind tunnel which is specially designed for aeolian studies. The technical details of the wind tunnel can be found in Zhang et al. (2014) and hence only the most relevant information is given here. Figure 1 shows the wind-tunnel configuration for the experiment. The working section of the tunnel is about 15 m long, with the first 6 m being the roughness-element section for generation of a turbulent boundary layer. The rest section is covered by a 40-grit sand paper to simulate a non-erodible sandy surface. One end of the piece of cloth is attached to a horizontal bar located 6 m downstream the roughness-element section and 0.7 m above the tunnel floor, and the other end is allowed to flutter freely.The cloth is a woven fabric (grammage $200 \, \mathrm{g \cdot m^{-2}}$) with size of 1 m in width and 1.5 m in length. The cloth size was empirically determined by a series of tests before the formal experiment, to satisfy the requirement on generating quasi-convective turbulence. Two sand trays [285 mm wide, 150 mm long and 13 mm deep, which have been tested as a suitable option for the study of aerodynamic entrainment (Li et al., 2020)] are placed 1.5 m downstream the end of the fluttering cloth. The trays filled with sand are mounted flush to the tunnel floor. The sand surface is smoothed before every test. Each tray is weighted before and after each test by an electronic balancer with a precision of 0.01 g in the range 5 kg, to determine the net mass loss of the tested surface. The anemometers, including the hotwire (1-D, fixed at 10 mm height and employed only in clear air condition) and the wind profiler ( combined by nine pitot tubes, and placed at the levels of 6.5, 10, 15, 30, 60, 120, 201, 351 and 501 mm), were located between the trays. The outer diameter of the pitot tubes of the wind profiler is 1 mm, and the inner diameter 0.5 mm. The wind profiler measures the profile of the mean flow speed with a sampling frequency of 1 Hz, while the hotwire anemometer measures turbulent fluctuations with a sampling frequency of 1000 Hz. An Irwin sensor is mounted on the central axis of the wind tunnel floor and locates upwind of the tray. Irwin sensors (Irwin, 1981) are omnidirectional devices for measuring the surface shear stress, which have been used successfully in a number of earlier studies (e.g., Wu and Stathopoulos, 1993; Walter et al., 2012). We manufactured the Irwin sensors used in this study ourselves according to

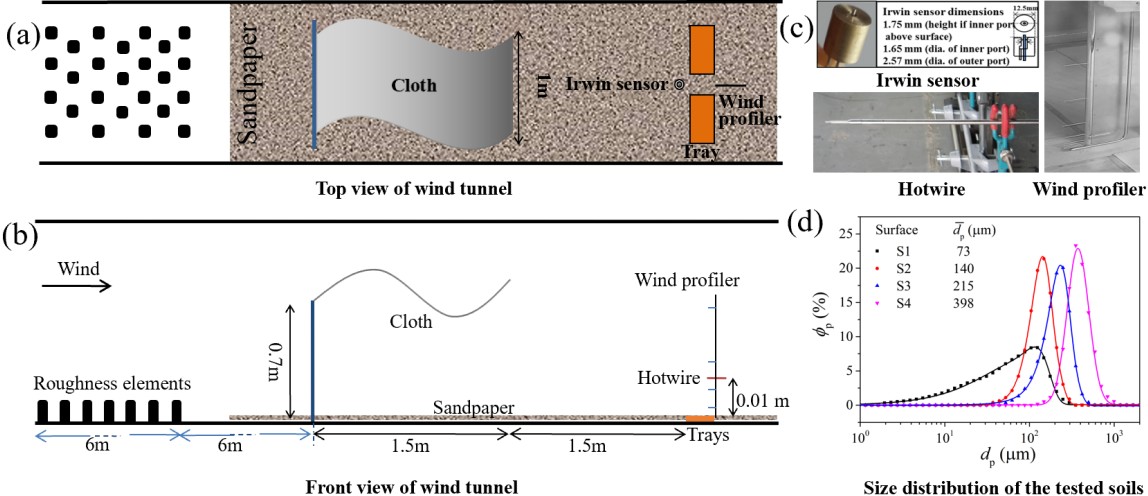

**Figure 1.** (a) Top view of the wind-tunnel configuration; (b) Side view of the wind-tunnel configuration, a piece of randomly fluttering cloth generates quasi-convective turbulence; (c) The probes employed. The Irwin sensor has a diameter of 12.5 mm. Its inner port has a diameter of 1.65 mm and is 1.75 mm in height, and its outer port diameter is 2.57 mm; (d) The size distributions of four tested soils, $\phi_p$ is volume fraction, and $d_p$ is particle diameter.

the dimensions given in Gillies et al. (2007), as shown in Fig. 1(c). The Irwin-sensor pressure differences are sampled at a frequency of 100 Hz using a scanivalve attached to a pressure transducer (ZOC33). Prior to the wind-tunnel experiment, the

Irwin sensors are calibrated in the wind-tunnel against the wind profiler for various flow conditions.

A key requirement for our experiment is to generate turbulence in the wind tunnel with characteristics similar to convective turbulence. In convective ABLs, large eddies develop due to the buoyancy production of turbulent kinetic energy. While horizontal velocity fluctuations are approximately Gaussian distributed, vertical velocity fluctuations are typically non-Gaussian with a positive skewness, resulting in a positive skewed probability distribution of surface shear stress. As already

discussed, convective turbulence is difficult to generate in wind-tunnel flows which are usually neutrally stratified. Here, we use a 'forced perturbation' technique to generate turbulence in the wind-tunnel flow, that mimics convective turbulence with energy containing large eddies and positively skewed velocity probability distribution function (PDF). Such turbulence is here as quasi-convective turbulence. Forced perturbation is achieved by using a piece of cloth which flutters (rapid and lightly swing) randomly in the wind-tunnel flow to produce small eddies, and flaps with longer period and greater amplitude to produce large

eddies, superposed on the background turbulence. Although quasi-convective turbulence is not the same as convective turbulence, the forced-perturbation method is both simple and efficient to remedy the critical deficit of wind-tunnel flows which lack of large eddies with skewed structures.

**Table 1.** Summary of wind-tunnel experiments

| Fan speed (×100 rpm) Particle size | Repetitions | | | | | | | | | |
|---|---|---|---|---|---|---|---|---|---|---|
| | 70 NP/WP | 75 NP/WP | 80 NP/WP | 85 NP/WP | 90 NP/WP | 95 NP/WP | 100 NP/WP | 105 NP/WP | 110 NP/WP | 120 NP/WP |
| O (no soil) | 1 | / | 1 | / | 1 | / | 1 | / | 1 | 1 |
| S1 (75 $\mu$m) | 5/3 | / | 5/5 | / | 5/5 | / | 5/5 | / | 5/5 | 3/5 |
| S2 (140 $\mu$m) | 5/5 | /5 | 5/5 | /4 | 5/5 | 3/3 | 3/ | / | / | / |
| S3 (215 $\mu$m) | 5/5 | / | 5/5 | /5 | 5/5 | / | 5/5 | 5/ | 6/ | / |
| S4 (398 $\mu$m) | 3/3 | / | 3/5 | / | 3/5 | / | 5/5 | / | 5/5 | 5/ |

Note: NP = no cloth; WP = with cloth. The tests of 0.5×1000 rpm are supplementary. In the case of large surface shear, the erodible surface (S2 and S3) rapidly appeared obvious surface-concave which could affect the test results. We therefore added several tests for low surface shear.

The wind tunnel is a blow tunnel, with the inlet flow speed controlled by a rotating fan. For our experiment, the fan speed is fixed for each run between 7000 rpm and 12000 rpm with interval of 1000 rpm, and the corresponding inlet free wind speed is between 7.7 m·s$^{-1}$ and 13.7 m·s$^{-1}$. We call the runs with forced perturbation WP-runs and those with no forced perturbation NP-runs. The entrainment rate are measured for various flow and turbulence combinations, as listed in Table 1. For each run, at least three successful repetitions are made and the measurement period for each repetition is 5 minutes. Four different soils are used in the experiment, labelled as S1-S4. The mean particle size of the four soils are respectively 75, 140, 215 and 398 $\mu$m. The particle size distributions are approximately log-normal, as shown in Fig. 1(d), measured by a Microtrac S3500 Laser Diffractometer (Microtrac, Montgomeryville, USA). We use NP70_S1 to denote the NP-run for fan speed 7000 rpm and soil S1 and name following this convention the other runs.

## 3 Results

### 3.1 Forced Perturbation

We first examine whether turbulence generated using the forced-perturbation technique has the desired features of convective turbulence. In Fig. 2, the characteristics of $V_{10\mathrm{mm}}$ (Horizontal flow velocity in height of 10 mm sensed by the hotwire anemometer) are compared between the NP70_O and WP70_O runs, including its time series, PDF and power spectrum. As $V_{10\mathrm{mm}}$ is measured using a one-dimensional hotwire, it is the resultant velocity of its horizontal component, $u_{10\mathrm{mm}}$, and vertical component, $w_{10\mathrm{mm}}$. As seen, the forced-perturbation technique effectively generates quasi-convective turbulence, as turbulence for the WP70_O run has an increased variance and a positive skewness, while turbulence for the NP70_O run is weaker and almost Gaussian distributed. It is shown that the effect of cloth not only enhances the average value of instantaneous wind speed, but also causes the probability of strong wind to increase in the distribution of instantaneous wind speed. While depending on the fluttering mode of the cloth, the quasi-convective turbulence has coherent structures as observed in

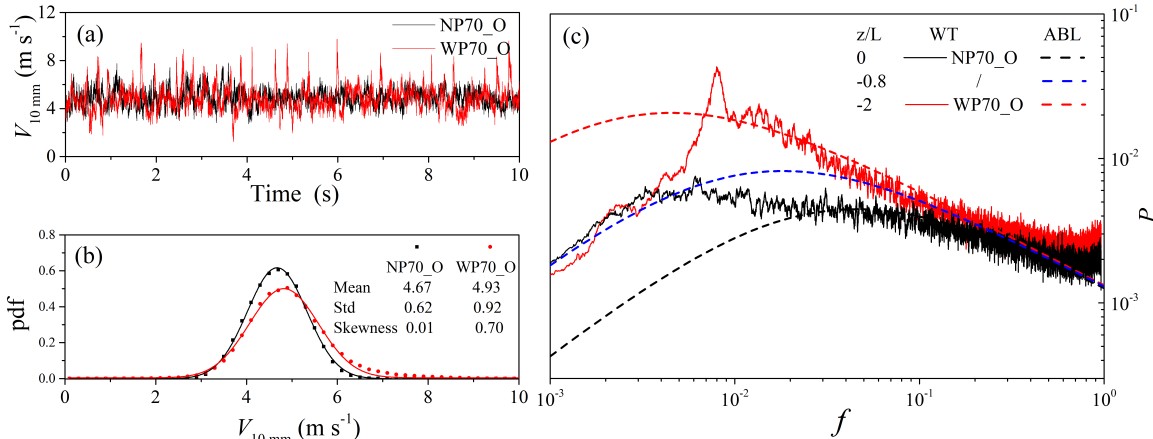

**Figure 2.** (a) A section of 10s of the $V_{10\mathrm{mm}}$ time series for the NP70_O and WP70_O runs; (b) probability density functions of $V_{10\mathrm{mm}}$ (estimated using a time series of 300 s); (c) normalized power spectra of $V_{10\mathrm{mm}}$ for WP70_O and NP70_O compared with field-observed power spectra in atmospheric boundary layer (Kaimal et al., 1972). $Z_{0\mathrm{WT}}$ is $1.33\times10^{-2}$ mm and $Z_{0\mathrm{ABL}}$ is set to 3.10 mm (Wiernga, 1993).

convective ABLs (Liu and Zheng, 2021). As our main interest is how convective eddies affects aeolian particle entrainment, we did not study the intrinsic mechanisms for how cloth induces large eddies.

In the MOST (Monin-Obukhov Similarity Theory) framework, $\zeta = z/L$, with $z$ being height and $L$ the Obukhov length, is used as a measure of ABL stability: the ABL is stable, neutral and unstable if $\zeta > 0$, $= 0$ and $< 0$, respectively. Kaimal et al. (1972) examined the characteristics of surface-layer turbulence using the MOST and found that the nondimensionalized power spectra of ABL quantities collapse to universal functions with $\zeta$ being the only parameter. They showed that as the ABL stability decreases, the inertial subrange extends to lower frequencies. Plotted in Fig. 2(c) are the normalized power spectra of $V_{10mm}$ for NP70 and WP70, denoted as $P_{\mathrm{NP70}}$ and $P_{\mathrm{WP70}}$, respectively. Following Kaimal et al. (1972), we express the normalized frequency as $f = nz/u$ (with $n$ being frequency) and normalized energy spectral density as

$$P = nS(n)\overline{u}_*^{-2}\phi_\epsilon^{-2/3} \tag{2}$$

with $S$ being the energy density per frequency and $\phi_\epsilon$ the MOST similarity function for the dissipation rate of turbulent kinetic energy

$$\phi_\epsilon = 1 + 0.5|\zeta|^{2/3} \tag{3}$$

following Kaimal and Finnigan (1994), our analysis does not involve the cases of $\zeta > 0$. For the wind-tunnel runs, $nS(n)$ is obtained by analyzing the horizontal wind velocity component measured by the hotwire, $\overline{u}_*$ is measured by the Irwin sensor and $\phi_\epsilon$ is calculated by using Eq. (3) with $\zeta = 0$ for the NP70_O run and $\zeta = -2$ for the WP70_O run. For comparison, the ABL velocity power spectra, denoted as $P_{\mathrm{ABL}}$, for three different stabilities $\zeta = 0$, -0.8 and -2 are plotted. The empirical form given

 by Kaimal et al. (1972)

$$P_{\text{ABL}} = 105f/(a + 33f)^{5/3} \tag{4}$$

is used, where $a$ is an empirical constant set to 1, 0.4 and 0.1 for $\zeta = 0$, -0.8 and -2, respectively. In Fig. 2(c), $P_{\text{ABL}} \times z_{0\text{WT}}/z_{0\text{ABL}}$ is plotted, where $z_{0\text{WT}}$ is the roughness length for the wind-tunnel flows and $z_{0\text{ABL}}$ that for the field ABL flows. The ratio $z_0/L$ is the MO number which was excluded in Kaimal et al. (1972). The exclusion is justified because the differences in $z_0$ in their data are not large. However, because $z_{0\text{WT}}$ (Table 2) is two orders of magnitude smaller than $z_{0\text{ABL}}$, the effect of $z_0/L$ needs to be considered and hence the mentioned multiplication is necessary.

Figure 2(c) reveals that $P_{\text{NP70}}$ and $P_{\text{WP70}}$ are almost the same in the (normalized) frequency range of $f < 3 \times 10^{-3}$, as turbulence in this frequency range is attributed to the upstream roughness elements. They are also the almost the same in the high frequency range of $f > 0.1$. In the energy containing range $3 \times 10^{-3} < f < 0.1$, $P_{\text{WP70}}$ shows much increased energy with respect to $P_{\text{NP70}}$, implying that the forced perturbation technique generated large eddies in the wind-tunnel flow.

Figure 2(c) also shows that it is generally difficult for the wind tunnel to reproduce the turbulence observed in the ABL. Clearly, compared with $P_{\text{ABL}}(\zeta = -0.8)$ and $P_{\text{ABL}}(\zeta = -2)$, $P_{\text{NP70}}$ lacks energy in the frequency range of $3 \times 10^{-3} \sim 1 \times 10^{-1}$. In contrast, power spectral density in this frequency range is substantially increased if forced perturbation is applied as a comparison of $P_{\text{WP70}}$ and $P_{\text{NP70}}$ reveals. It is seen that $P_{\text{WP70}}$ is fairly similar to $P_{\text{ABL}}(\zeta = -2)$, although it still lacks energy for $f < 10^{-2}$. In summary, Fig. 2 shows that the forced-perturbation technique is effective in generating quasi-convective turbulence which has a degree of similarity with ABL convective turbulence. This simple technique can be further optimized (e.g., by using a combination of fluttering cloths of different materials and different dimensions) to overcome the critical lack of convective eddies in wind-tunnel flows, which has so far seriously limited the usefulness and generalization of the wind-tunnel results.

## 3.2 Mean Wind Profile and Shear Stress

Figure 3 shows the mean wind profiles measured using the pitot tubes. For height $z$ smaller than 0.2 m, the mean wind profiles are for both NP runs and WP runs are approximately logarithmic. In the WP runs, the flow speed for $z > 0.2$ m is reduced due to the fluttering cloth which acts as a momentum sink. For z < 0.2 m in WP runs, the air flow seems to be accelerated ($\sim$ 0.5 $\text{m} \cdot \text{s}^{-1}$). It could be a wind tunnel artifact associated with a small degree of compression of the flow that was redirected beneath the cloth and between the confining walls. For a given fan speed, the fluttering cloth not only enhances the turbulent kinetic energy (Fig. 2) but also modifies the wind profile for $z < 0.2$ m, which will be proved by the following analysis.

Based on the MOST, the similarity relationship between the mean flow speed $\overline{U}$, can be expressed as

$$\overline{U}(z) = \frac{\overline{u}_*}{\kappa}[ln(\frac{z}{z_0}) + \Psi_{\text{m}}] \tag{5}$$

where $\kappa$ is the von Karmen constant, and

$$\Psi_{\text{m}} = \int_{z_0}^{z} (1 - \phi_{\text{m}}) dlnz \tag{6}$$

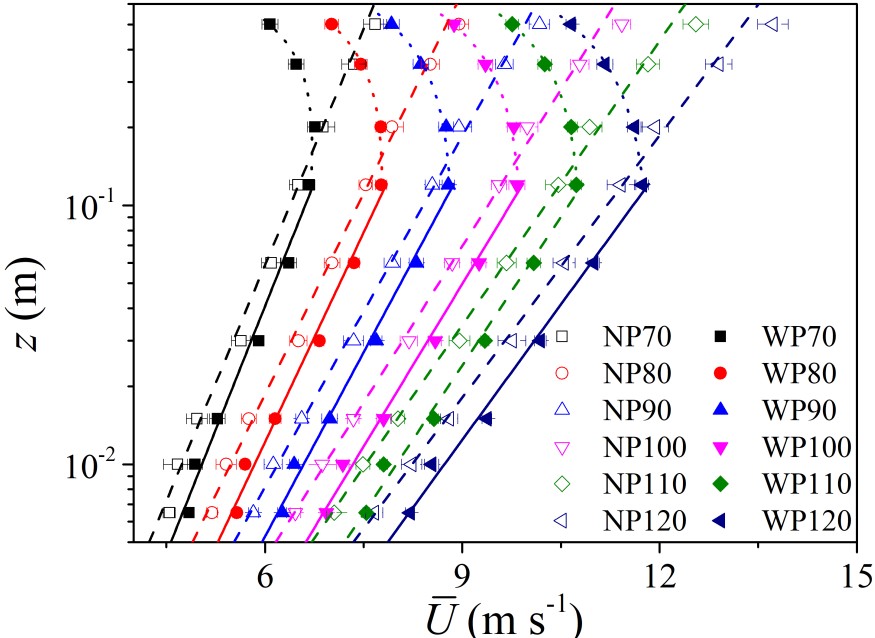

**Figure 3.** Mean flow speed profiles for different runs. For all cases, the wind tunnel floor is covered by sandpaper. The trays filled with particles and mounted flush to the tunnel floor does not affect the measured profiles.

and

$$\phi_{\mathrm{m}} = (1 - \gamma_{\mathrm{m}}\zeta)^{-1/4}, \zeta \le 0 \tag{7}$$

being the similarity function. For ABL flows, $\gamma_{\mathrm{m}} = 16$ is an empirical coefficient (Businger et al., 1971). For the NP runs, $\Psi_{\mathrm{m}} = 0$ is assumed. By fitting Eq. (5) to $\overline{U}$ measured at $z < 0.2$ m, we estimate $\overline{u}_*$ and $z_0$. The shear stress $\overline{\tau} = \rho\overline{u}_*^2$ (here, air density

$\rho = 1.2$ kg $\cdot$ m$^{-3}$) is then used to calibrate the shear stress measured using the Irwin sensor, $\overline{\tau}_{\mathrm{Irwin}}$. But if $\Psi_{\mathrm{m}}$ is set to 0 for the case of WP, the obtained $u_*$ obviously diverge from the data of Irwin sensor. Only when a non-zero $\Psi_{\mathrm{m}}$ is considered, the deduced $u_*$ agrees with the data of Irwin sensor (Table 2). That is why we state that the wind profile for $z < 0.2$ m modifies to one more similar to wind profile in convective ABL. As both $\gamma_m$ and $L$ are unknown for the quasi-convective turbulent flows, it is sensible to write Eq. (7) as

$$\phi_{\mathrm{m}} = (1 - \eta_{\mathrm{m}}z)^{-1/4} \tag{8}$$

with $\eta_{\mathrm{m}} \sim \gamma_{\mathrm{m}}/L$. By combining the $\overline{U}$ measurements at $z < 0.2$ m and the shear stress measured using the Irwin sensors for the WP runs, $\eta_{\mathrm{m}}$ can be estimated. The results are summarized in Table 2.

**Table 2.** Friction velocity $\overline{u}_*$ and roughness length $z_0$ estimated for runs with and with no forced perturbation and different wind-tunnel fan speeds ($R^2 = 99\%$). The Irwin sensor is calibrated based on the data of wind profiles under the NP condition.

| Fan speed | $U_{501mm}$ | NP | | WP | | | Irwin |
| | | Profile($\Psi_m = 0$) | | Profile($\Psi_m \neq 0$) | | | |
| ($\times 100$ rpm) | (m·s$^{-1}$) | $u_*$ (m·s$^{-1}$) | $z_0$ (mm) | $u_*$ (m·s$^{-1}$) | $z_0$ (mm) | $\eta_m$ (mm$^{-1}$) | $u_*$ (m·s$^{-1}$) |
|---|---|---|---|---|---|---|---|
| 70 | 7.68 | 0.29±0.0087 | 0.0133±0.0044 | 0.31±0.0077 | 0.0150±0.0034 | -0.0229 | 0.32±0.0128 |
| 75 | 8.32 | / | / | 0.34±0.0062 | 0.0153±0.0025 | -0.0213 | 0.34±0.0064 |
| 80 | 8.95 | 0.34±0.0068 | 0.0146±0.0024 | 0.36±0.0067 | 0.0159±0.0016 | -0.0200 | 0.37±0.0101 |
| 85 | 9.56 | / | / | 0.39±0.0051 | 0.0167±0.0013 | -0.0188 | 0.39±0.0069 |
| 90 | 10.18 | 0.38±0.0083 | 0.0159±0.0020 | 0.41±0.0087 | 0.0156±0.0022 | -0.0178 | 0.42±0.0080 |
| 95 | 10.80 | / | / | 0.44±0.0038 | 0.0165±0.0007 | -0.0168 | 0.44±0.0031 |
| 100 | 11.42 | 0.43±0.0106 | 0.0164±0.0024 | 0.46±0.0083 | 0.0166±0.0016 | -0.0160 | 0.47±0.0109 |
| 110 | 12.55 | 0.47±0.0204 | 0.0175±0.0043 | 0.50±0.0067 | 0.0165±0.0010 | -0.0146 | 0.51±0.0077 |
| 120 | 13.72 | 0.51±0.0180 | 0.0166±0.0035 | 0.55±0.0033 | 0.0162±0.0008 | -0.0133 | 0.55±0.0085 |

Table 2 shows that forced perturbation leads to an increased $\overline{u}_*$, corresponding to an increase of $\overline{\tau}$ by about 22% at fan speed 7000 rpm and about 16% at fan speed 12000 rpm. As pointed out in several earlier studies (Klose and Shao, 2012; Li et al., 2020; Shao et al., 2020), we emphasis again that surface shear stress $\tau$ is a stochastic variable, which satisfies a probability distribution function $p(\tau)$. To facilitate discussions, we explicitly write

$$\tau = \overline{\tau} + \tau^{'} \tag{9}$$

with $\tau^{'}$ being the perturbation of $\tau$.

### 3.3 Aeolian Particle Entrainment in Quasi-convective Turbulence

The entrainment rate of sand and dust particles $F$, is estimated from the mass loss of the trays as

$$F = \frac{1}{I} \sum_{i=1}^{i=I} \frac{\Delta m_i}{A \Delta T_i} \tag{10}$$

where $\Delta m_i$ is the net mass loss (integrated over $\Delta T_i$) from the tray during the ith run with run time $\Delta T_i$, $A$ is the tray surface area and $I$ is the number of repetitions. Figure 4 shows that the entrainment rates of the various particle-size groups measured in the NP and WP runs. It is seen that for all four soils, for given $\overline{\tau}$, the entrainment rates for the WP runs are substantially larger than that for the NP runs. This result suggests that in addition to the mean surface shear stress $\overline{\tau}$, the surface shear stress perturbations $\tau^{'}$, significantly influence the entrainment rate. It shows that the slight increase of $\overline{\tau}$ in quasi-convective

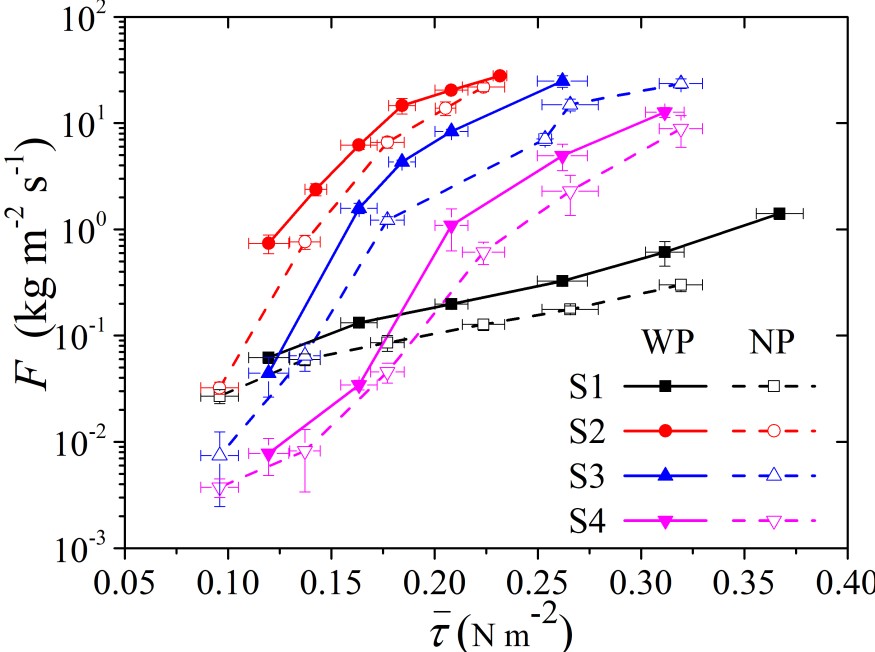

**Figure 4.** Entrainment rate of four different soils observed in the NP and WP runs. Hollow symbols represent the NP runs and solid symbols for the WP runs.

conditions is not sufficient alone to explain the measured differences in the entrainment rates of the four soils. It implies that the perturbations of the shear stress, $\tau'$ are also responsible for a part of the differences in the entrainment rates. As $\tau'$ is related to the structure of boundary-layer turbulence, it can be said that the structure of boundary-layer turbulence also influences the

200 entrainment rate: for a given mean surface shear stress, convective turbulence is more efficient in lifting particles from the surface into the air. This finding is consistent with the observations of the Japan-Australian Dust Experiment (JADE, Shao et al., 2020), i.e., aeolian sand transport and dust emission are much more intensive in convective ABLs than in stable ABLs.

Using the $\tau$ measurements of the Irwin sensor, we estimate the PDF of $\tau$, $p(\tau)$. Klose et al. (2014) and Shao et al. (2020) suggested that $p(\tau)$ is approximately Weibull distributed and positively skewed:

$$p(\tau) = \frac{K}{\lambda}(\frac{\tau}{\lambda})^{K-1}\exp\left[-(\frac{\tau}{\lambda})^{K}\right]$$
(11)

where $K$ is shape parameter, and $\lambda$ is scaling parameter.

Figure 5 shows as example $p(\tau)$ for the NP and WP runs for fan speed 7000, 9000 and 12000 rpm. As seen, the forced-perturbation results in significantly different PDF of $\tau$ by slightly increasing $\overline{\tau}$ and clearly increasing the probability of large $\tau$. It is this increase in the probability of large $\tau$, that explains the differences between the $F \sim \overline{\tau}$ dependency between the NP

and WP runs seen in Fig. 4.

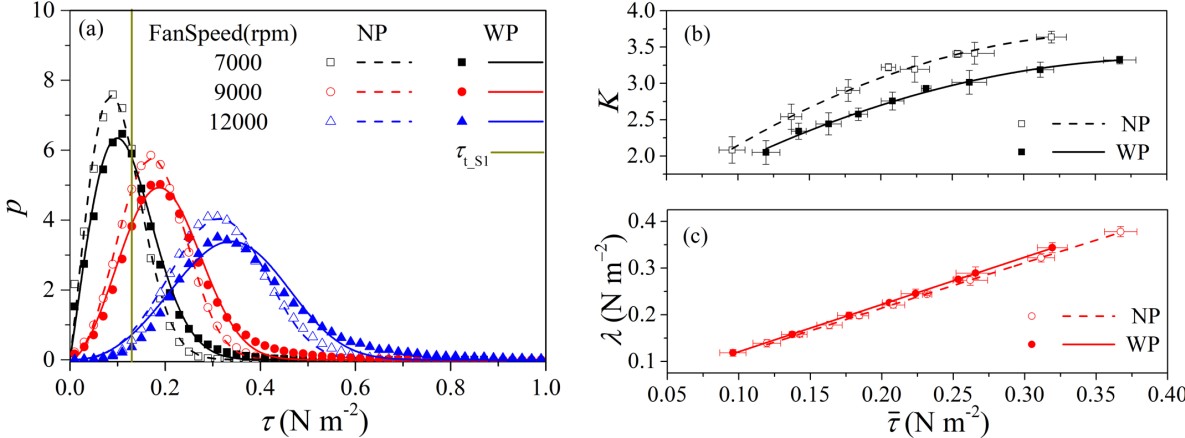

**Figure 5.** (a) Probability density function of surface shear stress $\tau$ for NP and WP runs with fan speed 7000, 9000 and 12000 rpm. The dashed gray line marks the threshold shear stress for S1 ($\tau_{t\_S1}$). The symbols are the results from the Irwin sensors and curves the Weibull distributions. By fitting Eq. (11) to the respective data of the various runs, the corresponding shape parameter $K$ and scaling parameter $\lambda$ are estimated and plotted against the mean shear stress $\bar{\tau}$ in (b) and (c), respectively. The curves are polynomial fits, (b) black solid line, $K = -15.10\tau^2 + 12.28\tau + 0.85$, black dashed line, $K = -21.15\tau^2 + 15.65\tau + 0.79$; (c) red solid line, $\lambda = 1.01\tau + 0.02$, red dashed line, $\lambda = 0.97\tau + 0.02$.

;

The structure of ABL turbulence, reflected here in $p(\tau)$, significantly influences the sand and dust entrainment and saltation fluxes, because this depends non-linearly on $\tau$. As explained in (Shao, 2008, sect 6.12.4), for a given particle size, the entrainment rate can be expressed as

$$F = \gamma\sqrt{\tau/\rho}(\tau - \tau_t) \tag{12}$$

with $\gamma$ being an empirical efficiency parameter for particle entrainment, and $\tau_t$ is the threshold shear stress for particle entrainment. The threshold is in general a stochastic variable (Raffaele et al., 2016; Liu et al., 2018), but for simplicity, we assume it is constant for given particle size. To account for the fluctuations of $\tau$, we estimate

$$F = \int_{\tau_t}^{\infty} \gamma\sqrt{\tau/\rho}(\tau - \tau_t)p(\tau)d\tau \tag{13}$$

for the four soils S1, S2, S3 and S4 tested, $\tau_t$ is estimated to be 0.13, 0.27, 0.31 and 0.37 N·m$^{-2}$, respectively. Figure 6 shows that Eq. (12) fits well to the measurements for all runs. Equation (12) reveals that convective turbulence may influence both the efficiency parameter $\gamma$, and the statistical behavior of the term, $\int_{\tau_t}^{\infty} \gamma\sqrt{\tau/\rho}(\tau - \tau_t)p(\tau)d\tau$. To facilitate discussion, we rewrite Eq. (12) as

$$F = \bar{\gamma}\int_{\tau_t}^{\infty} \sqrt{\tau/\rho}(\tau - \tau_t)p(\tau)d\tau \tag{14}$$

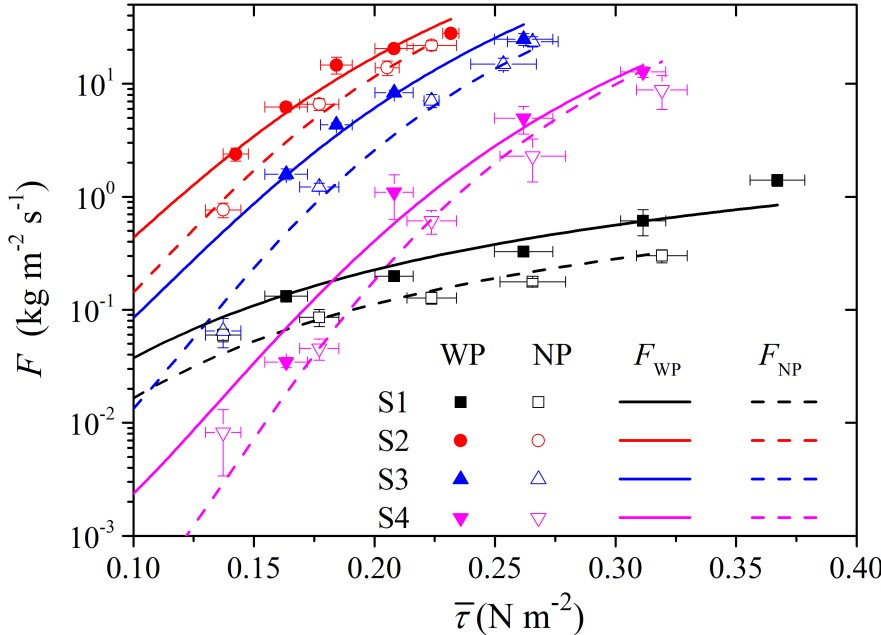

**Figure 6.** Estimated entrainment rates with and without forced perturbation. The dots are experimental data and lines derive from Eq. (12).

A comparison of $\bar{\gamma}$ for NP and WP runs are shown in Table 3. It is seen, that $\bar{\gamma}$ is significantly increased for the WP runs. The

low value of $\bar{\gamma}$ for S1 is supposed to be caused by the wide distributed particle size, leading to an increased mean threshold friction velocity when wind speed increases, which means constant threshold is not suitable in this situation. However, this is beyond the scope of this work, which mainly focus on the comparison of the WP and NP conditions. Thus, we estimate the ratio $\sigma_F$ defined as

$$\sigma_F = \int\limits_{\tau_t}^{\infty} \sqrt{\tau/\rho}(\tau - \tau_t)p_{WP}(\tau)d\tau \bigg/ \int\limits_{\tau_t}^{\infty} \sqrt{\tau/\rho}(\tau - \tau_t)p_{NP}(\tau)d\tau \tag{15}$$

Figure 7 shows the relationship between $\sigma_F$ and excess surface shear stress $(\bar{\tau} - \tau_t)$. A negative exponential law appears to exist. For the conditions with $\bar{\tau} > \tau_t$ [corresponding to the continuous entrainment defined by Li et al. (2020)], $\sigma_F$ is close to one, indicating relatively small influence from the quasi-convective turbulence. But for the conditions with $\bar{\tau} < \tau_t$, $\sigma_F$ significantly increase with decreasing $\bar{\tau} - \tau_t$, reaching up to 6 at $\bar{\tau} - \tau_t = -0.25 \ \mathrm{N \cdot m^{-2}}$, indicating that the influence of convective turbulence is significant [corresponding to the intermittent entrainment defined by Li et al. (2020)]. We can

thus conclude that convective turbulence may significantly enhance dust entrainment by altering how shear stress acts on the surface, especially for the cases of intermittent entrainment when the mean shear stress is below the threshold. Considering that

**Table 3.** Threshold shear stress $\tau_t$ and empirical parameter $\gamma$ for test surfaces.

| soil type | $\tau_t$ (N·m$^{-2}$) | WP | | NP | | $\frac{\overline{\gamma_{NP}}}{\overline{\gamma_{WP}}}$ |
|---|---|---|---|---|---|---|
| | | $\overline{\gamma_{WP}}$(m$^{-2}\cdot$s$^2$) | $R^2_{WP}$ | $\overline{\gamma_{NP}}$(m$^{-2}\cdot$s$^2$) | $R^2_{NP}$ | |
| S1 | 0.13 | 7.12 | 0.89 | 3.30 | 0.86 | 1.96 |
| S2 | 0.27 | 4919.84 | 0.95 | 3605.29 | 0.99 | 1.29 |
| S3 | 0.31 | 4675.62 | 0.97 | 2521.74 | 0.98 | 1.97 |
| S4 | 0.37 | 1996.39 | 0.95 | 1713.19 | 0.95 | 1.34 |

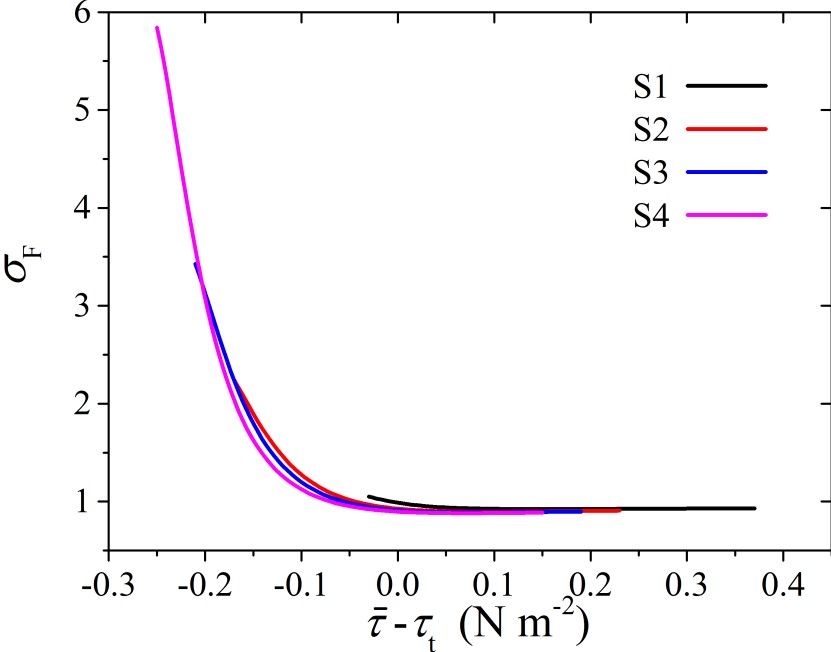

**Figure 7.** Relationship between $\sigma_F$ and excess surface shear stress $(\overline{\tau} - \tau_t)$.

the lower-than-threshold wind conditions may have a high temporal weight in natural conditions, we believe that this effect deserves particular attention in dust emission schemes.

## 4 Conclusions

We carried out wind-tunnel experiments and studied the influences of turbulence structure on aerodynamic entrainment of sand and dust particles. We considered $\tau$ to be a stochastic variable and showed that the probability distribution of $\tau$ (i.e., $p(\tau)$), in addition to the mean surface shear stress $\overline{\tau}$, has a significant impact on aeolian fluxes, as the entrainment rate of sand and dust particles depends non-linearly on surface shear stress, $\tau$. Because the fluctuations of $\tau$ are closely related to the structure of ABL turbulence, aeolian fluxes in ABLs of different stabilities can be substantially different even if $\overline{\tau}$ is the same. The wind-tunnel experiments provided direct data which show that ABL convective large eddies are of particular importance to the entrainment of sand and dust particle, as they not only increase the mean shear stress on the surface, but also increase the probability for instantaneous shear stress to exceed the threshold, leading to intermittent entrainment.

Wind-tunnel flows are normally neutrally stratified and do not contain large eddies similar to those in convective ABLs. By examining the power spectra of turbulence for the NP runs, we showed that wind-tunnel turbulence lacks energy containing eddies even compared with ABL flows in neutral conditions. Although advanced techniques for measuring turbulence structure and intensity associated with sediment entrainment and transport have been developed in the past few decades, some focusing on the effects of gusting wind (e.g., Li and McKenna Neuman, 2012; Li and Neuman, 2014), few studies have been done focusing on the effects of ABL convective turbulence. Also, means for generating large-scale eddies (not necessarily convective large eddies) in wind tunnel simulations for wind engineering applications, have not been widely used in aeolian studies due to their complexity and high operational cost. We showed that the deployment of a piece of fluttering cloth is both simple and effective in generating quasi-convective turbulence. By comparing the power spectra of turbulence for the WP runs and NP runs, we found that the energy density in the eddy energy containing range is substantially increased. The energy spectrum of quasi-convective eddies agrees reasonably well with that of turbulence in unstable ABL with $z/L$ = -2. Although the employment of the forced-perturbation technique did not fully reproduce the energy spectra of convective turbulence, this simple technique can be further developed and optimized to obtain the desired turbulent features and overcome a vital limitation to aeolian wind-tunnel experiments.

By comparing the WP runs and NP runs, we found that quasi-convective turbulence increases the mean value (just as in convective ABLs) as well as the variance and skewness of the surface shear stress, all contributing to the entrainment of sand and dust particles. For a given mean shear stress, the entrainment rate for the WP runs is substantially higher than for the NP runs, i.e., convective turbulence is more effective than neutral turbulence in entraining particles into the atmosphere. The enhancing effect is greatest at low wind speeds around threshold and when transport is intermittent, and becomes relatively weaker when the mean wind speed is strong enough and dominates over the fluctuations.

The findings of this study obtained through wind-tunnel observations are consistent with the results of Shao et al. (2020). The latter authors showed based on field observations that the PDF of $u_*$ can influence the magnitude of saltation flux, $Q$. With fixed $u_*$ mean, a larger $u_*$ variance corresponds to a larger $Q$. Unstable ABL has in general larger $u_*$ variances which generate stronger saltation bombardment for dust emission, and produce the emission of finer dust particles, and saltation in unstable ABLs is generally more fully developed, leading stronger saltation bombardment. In a more recent study, Yin et al.

(2021) demonstrated using large-eddy simulations that also dust deposition is strongly affected by the structure of turbulence. Together with the earlier studies of Shao (2008), Klose and Shao (2012), Klose et al. (2014), Li et al. (2020), Khalfallah et al. (2020), Shao et al. (2020), and Yin et al. (2021), we have shown the critical importance of taking into consideration of turbulence structure in aeolian studies, and have partly quantified the impact of turbulence on sand and dust entrainment, dust emission, saltation fluxes and dust deposition.

*Data availability.* Data is available from Dr. Zhang (zhang-j@lzu.edu.cn) or Dr. Li (liguang@lzu.edu.cn) on requests.

*Author contributions.* YS and JZ conceived and designed the wind tunnel experiment; JZ, GL, and LS carried out the experiment, performed the data analyses, and prepared the first draft; YS and NH organized this study and contributed to its conceptualization, discussions, and finalization of the paper.

*Competing interests.* The authors declare that they have no conflict of interest.

*Acknowledgements.* This work is supported by the National Natural Science Foundation of China (41931179, 42006187), the Major Science and Technology Project of Gansu Province (21ZD4FA010), the Second Tibetan Plateau Scientific Expedition and Research Program (2019QZKK020611), and the Fundamental Research Funds for the Central Universities (lzujbky-2020-cd06).

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
