# Peer review of "Impact of Turbulence on Aeolian Particle Entrainment: Results from Wind-tunnel Experiment"

_Atmospheric Chemistry and Physics, 2022_

## Referee Comment (RC1)

This paper aims to investigate the effects of atmospheric turbulence on sediment emission rates. This topic is a central issue in recent investigations by the community with various papers suggesting important effects of turbulence on dust fluxes and their size distribution.

This paper contains two major elements: it implements a simple technique to generate quasi-convective turbulence in a wind tunnel. This is an important advance because the lack of wind tunnel experiments on sediment movement has so far been carried out under neutral or near-neutral conditions which do not allow the study of the role of large eddies on wind erosion. The second major point, well illustrated by Figure 4, is that the results show that the sediment emission rates are higher during quasi-convective turbulence situations. These are significant contributions that deserve a publication of this paper.

Nevertheless, in its form, some parts need to be better detailed to improve understanding or to avoid unanswered questions. Most figures need to be enlarged.

Comments:

Lines 17-18: I agree that "it has been shown in numerous studies that equation 1 is valid in general" but some papers have also challenged its validity (e.g. Martin & Kok, Science Advances, 2017; Andreotti, J. Fluid Mech., 2004). It is not the purpose of this paper to discuss this but the wording used could suggest that this debate does not exist.

Lines 74-75: I understand the first part of the sentence, which seems sufficient in itself. What more do you mean by "by comparing the shear stresses measured by the two devices"?

Lines 92-94: The description of the soils is really limited. On reading, one gets the impression that only the average particle size differs and this is not sufficient to then understand why $\bar{\gamma}$ are so different in table 3.

Line 141: the end of the sentence is not clear for me. When looking at figure2, the wind profile is not significantly modified for z<0.2m. The authors should be more explicit.

Table 2: This table needs to be strongly completed and better discussed. I do not retrieve the 25% and 15% difference in $\bar{\tau}$ for fan speeds 7000 rpm and 12000 rpm, respectively, as mentioned line 157, when comparing $u_*$ from the NP and WP profiles. Are the $u_*$ from Irwin sensors those obtained for the WP conditions. This should be clearer in the table (move WP to be centered). Moreover, there are 4 Irwin sensors in the wind tunnel: how do the authors use them? Is the reported data an average of the four sensors? If so, give the standard deviation. More generally, give the standard deviations since there are several repetitions of each experiment. Add the emission rate for each experimental condition and the associated standard deviation.

Why does $z_0$ change significantly for NP experiments: it should be constant unless there is an additive saltation roughness but the WP experiments have an almost constant $z_0$ which rules out this assumption. How accurate is the recovery of $u_*$ and $z_0$ from the wind profile? Does this accuracy depend on the regime (i.e. wind speed)?

Figure 4 is the key figure of the paper. It shows that the slight increase of $\bar{\tau}$ in quasi-convective conditions is not sufficient alone to explain the measured differences in the emission rates of the four soils. It implies that the perturbations of the shear stress, $\tau'$, are also responsible for a part of the differences in the emission rates. This is perfectly clear.

However, the lack of information on the different soils (S1 to S4) complicates the understanding of why the emission rates of the different soils as reported in figure 4 do not follow the order of $\tau_t$ (as shown in table 3). According to equation 12, the only explanation for that is that the different soils have different values of $\bar{\gamma}$ (as suggested by table 3 for NP experiments) but reasons or at least hypotheses allowing to understand this should be given. Furthermore, what could be the possible explanation for such large differences in $\bar{\gamma}$ observed for S1 and S4 but not for S2 and S3 between the two sets of experiments? I can understand that $\bar{\gamma}$ is affected by quasi-convective turbulence but it is difficult to understand why it would affect the four soils so differently.

The number given in Table 3 are very precise with two significant digits for all parameters but we have no idea of the uncertainties and how are significant the differences in $\bar{\gamma}$.

Moreover, the quality of the fits to equation 12 is not given while looking at figure 6 (and despite its small size), these fits seem worse for S1 and S4 than for S2 and S3.

Lines 200 -201: Does this suggest that the impact of convective turbulence should be rather limited on the total dust emission budget?

---

## Referee Comment (RC3)

This paper uses wind tunnel simulation and modelling to investigate the response of the sediment entrainment rate to large scale, quasi-convective eddies in boundary-layer air flows. This is a very important topic of interest within the aeolian research community, such that this paper is well aligned with current research needs.

With moderate revision, particularly the inclusion of a great deal of missing information, it may be suitable for publication.

Minor corrections:

**Title:** Define the term 'dust' in the context of this study. You should be clear that you actually do not measure/profile the aerosol concentration of the airflow in this study, so the title is misleading. Perhaps you should say 'silt' instead, or drop it altogether.

**Line 20:** Circular argument or poor wording, "turbulent flow is also turbulent"

**Lines 27-28:** Poor wording - "impact" (geophysical process associated with saltation) should be "effect". In addition, "flow is turbulent, and aeolian processes also." Aeolian processes are affected by turbulence but are **stochastic**.

**Line 74:** "scan valve" perhaps should read "scanivalve"

**Lines 84-84:** Define what you mean by 'flutter" versus "flap".

**Line 80:** Define what you mean by "positive skewness" with regard to vertical velocity fluctuation. Does positive refer to eddies bursting up away from the bed (+ve w) and negative to eddies sinking down toward the bed (-ve w)? This section is a bit vague.

**Line 89:** The fan speed in rpm is meaningless to the reader. Provide free stream velocities for WP instead.

**Line 93:** Either provide the standard deviations to indicated the sorting for each test sediment or a figure showing the full distribution of particle diameter within each sample. This is important as most test beds will undergo some degree of grain scale armouring when subjected to an air flow and this is likely to be strongly affected by $\tau'$.

**Figures** Much too small and almost impossible to read.

**Symbols** A number of mathematical symbols appear in the manuscript well before they are defined and sometimes they are either not defined at all or the symbol has an inconsistent typeface.

**Line 165** Specifically you mean the **net** mass loss (integrated over time).

**Lines 163 and 170** "Emission rate" interchanged with the "entrainment rate". The former is usually used when talking about dust and the later with the mass transport of sand. This may

confuse some readers. Since you are not measuring the dust flux (aerosol concentration) then if might be best to just say "entrainment rate" everywhere.

Substantive comments:

**Figure 1 and Section 2.** This is the weakest section of the paper with a great deal of missing information and weak justification of the experimental design.

Flapping cloth:

i) This is not a novel solution for creating turbulence in wind tunnels. Provide one or more references to previous work.
ii) Details of the flapping cloth are missing. What type and weight of fabric? Width is given but more importantly, how long was the sheet? What was the wavelength, amplitude and frequency of its oscillation?
iii) Rationale is not provided for the placement of the cloth relative to the test surface. Is the elevation scaled with the length of the cloth or distance from the leading edge of the sample tray?
iv) Spires are much more commonly used to generate large scale eddies in wind tunnel simulation work and these are typically placed upwind of the roughness element array. There is also a large literature on the effect of spires on turbulent flow and shear. Why did you elect to use a flapping cloth over spires?

Pitot tubes:
i) Provide dimensions. Pitot tubes come in a range of sizes, while large tubes places in a fixed vertical array can initiate flow stagnation – bluff body effect.
ii) What did you sample the pressure difference with? State precision and sampling time.

Measurement of the mass transport rate:
i) One of the greatest challenges with accuracy in measuring mass in the lab using an electronic balance is drift associated with changing air pressure in the room. So it is possible that any instantaneous fluctuation in F is the a consequence of both the change in mass of particles in the tray and the pressure perturbation in the flow. This would be particularly exacerbated by the presence of large scale eddies. Please explain in detail how you accounted for this in your analysis. Was the total mass loss obtained from measurements in the absence of an airflow or did you average the instantaneous mass transport rate throughout the test?

Hot wire anemometer and Irwin sensors:
ii) Please state in line 70 that the instrument only samples in 1D
iii) Was the wire ruggedized to withstand particle impact during saltation? If so, how did this affect sensitivity (time constant and precision)?

iv)      Please provide exact dimensions of the Irwin sensor rather than a citation. The height of the central port does vary from study to study and is important to know relative to the roughness of the bed surface. Describe also the roughness of the bed surface upwind of the sample trays. Was it smooth or roughened to give a suitable aerodynamic roughness matched to that of the sand in the tray?

**Section 2.1 Results**

**Table 1**
    i)       Why so many missing experiments? Indeed, there are so few experiments at the 0.5 X1000 rpm increment that it might be better to exclude these data altogether.  Add the freestream velocity values for the NP experiments, as rpm will make little sense to the reader.
    ii)      Describe the bed surface composition for your control runs without a tray/soil.

**Figure 3**

    i)       I am not clear on the surface/sediment texture characteristics pertaining to these data. Have the data for all textures been lumped together?
    ii)      The NP vertical profiles of the total velocity look great – a deep well-structured boundary-layer flow following the law of the wall. The WP profiles also look very good, but I think even more might be inferred from them than what is provided. There is indeed a high degree of flow stagnation in the upper part of the profile above 20 cm associated with partitioning of momentum to the flapping cloth turbulent but also turbulent energy dissipation within the wake flow downwind.  However, little is said about the proportionate acceleration of the air flow (~0.5 m/s) at all levels below 20 cm in all WP experiments. This is unexpected and could be a wind tunnel artifact associated with a small degree of compression of the flow that was redirected beneath the cloth and between the confining walls.
    iii)     It is really unfortunate in this study that the vertical component of the total wind speed was not isolated and sampled, as required for analysis of the eddy structure. As first identified by Thom (1975) for truly unstable conditions with convection, mechanical effects (roughness) dominate the near surface flow, but in moving away from the surface the eddies are stretched vertically and the momentum flux is enhanced. Since the vertical velocity is generally one to two orders of magnitude smaller than the horizontal component, the total wind speed as sampled in this study cannot be particularly sensitive to the eddy perturbation that was intentionally created.

**Table 2**

    i)    Again, I am not clear on whether the data have been averaged across all sediment textures or this is just one example for a specific texture.
    ii)   Why is the threshold friction velocity also not reported here?

iii) Over what elevations were the wind speed data used to calculate u*? If the boundary layer depth is taken to be 20 cm for the WP experiments and the inner constant stress region 15-20% of that, then your calculation might only be based on 4 data points. That is, the shear stress values for the WP experiments would carry greater error than those for the NP experiments. How does this uncertainty compare with the roughly 7% increase in u* associated with introduction of the flapping cloth?

iv) Under what conditions were the Irwin sensors calibrated – NP or WP? This has a large bearing on an explanation for why the Irwin sensor u* values sit slightly higher than for the WP data.

**Section 2.1.3**

**Line 171**

"Convective turbulence is much more efficient in lifting particles into the air". Caution should be exercised in making such a broad statement. This may well be true for very light particles (aerosols) entering suspension, given a settling velocity that is lower than the vertical velocity at which eddies burst away from the bed surface, but not so for very large sand particles. Once again, it is unfortunate that you can only infer such effects because you don't have the vertical component measurements.

**Figures 4 and 5**

There is no question that positive skewness in the instantaneous bed shear stress should increase the particle entrainment rate at low values near threshold, but the fact that the near surface flow is accelerated (especially at lower elevations) by insertion of the flapping cloth is likely to be equally if not more important in this particular study.

**Conclusions**

Logic dictates and your study shows that intermittent generation of large shear stresses on the bed surface enhances the entrainment of sand and silt sized particles. This should be qualified by some form of statement to indicate that this effect is greatest at low wind speeds around threshold and when transport is intermittent. In really strong winds mechanical or impact entrainment dominates, while the particle borne stress largely determines the transport under saturated conditions.

"we showed that wind-tunnel turbulence lacks energy containing eddies even compared with ABL flows in neutral conditions, highlighting the deficiency of traditional wind-tunnel experiments on aeolian studies". Again, this is a bit of an **overstatement**. Most traditional wind tunnel studies have modelled saturated flows (see above) where the particle cloud itself is so

dense that it alters the turbulence structure. Since the development of fast response pressure transmitters, cross-wire probes, and laser Doppler anemometers more than 2 decades ago, aeolian researchers working in wind tunnels have indeed provided detailed measurements of the turbulence structure and intensity associated with sediment entrainment and transport while also investigating the effects of wind gusting (e.g. Li and McKenna Neuman 2012, 2014 etc). Similarly, there are many means by which we can and do generate large scale eddies in wind tunnel simulation and these are widely practiced in engineering applications, particularly in investigations of wind loading on urban structures.

---

## Author Comment (AC2)

This paper contains two major elements: it implements a simple technique to generate quasi-convective turbulence in a wind tunnel. This is an important advance because the lack of wind tunnel experiments on sediment movement has so far been carried out under neutral or near-neutral conditions which do not allow the study of the role of large eddies on wind erosion. The second major point, well illustrated by Figure 4, is that the results show that the sediment emission rates are higher during quasi-convective turbulence situations. These are significant contributions that deserve a publication of this paper.

Nevertheless, in its form, some parts need to be better detailed to improve understanding or to avoid unanswered questions. Most figures need to be enlarged.

**Response**: we are most grateful to Prof. Gilles Bergametti for his encouraging comments and constructive suggestions. The Referee pointed out the essential contributions of this work to elucidate the effects of atmospheric turbulence on sediment emission. The suggestions will be considered and the relevant text and figures will be modified accordingly.

Comments:

Lines 17-18: I agree that "it has been shown in numerous studies that equation 1 is valid in general" but some papers have also challenged its validity (e.g. Martin & Kok, Science Advances, 2017; Andreotti, J. Fluid Mech., 2004). It is not the purpose of this paper to discuss this but the wording used could suggest that this debate does not exist.

**Response**: thanks for the comment. We will modify the sentences to make relevant statements more precise.

However, it should be note that the argument of the linear-relationship between saltation flux $Q$ and shear stress $\tau$ (or $u_*^2$) is confined to saturated aeolian flow condition (or steady-state saltation condition) with splash-dominated particle entrainment. The majority of the existing measurements do not support the $Q \sim \tau$ linear relationship. In our study, aerodynamic entrainment is dominant and Eq. 1 should be reasonable to be used.

Lines 74-75: I understand the first part of the sentence, which seems sufficient in itself. What more do you mean by "by comparing the shear stresses measured by the two devices"?

**Response:** indeed, the latter part of this sentence is redundant. We will delete it in the revision.

Lines 92-94: The description of the soils is really limited. On reading, one gets the impression that only the average particle size differs and this is not sufficient to then understand why $\bar{\gamma}$ are so different in table 3.

**Response:** sorry for this negligence on the description of soil. We actually measure the size distribution of the employed four soils (as shown in follow) by using a Microtrac S3500 Laser Diffractometer (Microtrac, Montgomeryville, PA, USA). We will add this information to help for well understanding of reverent results. However, due to the wet method used at that time, these results don't necessarily reflect the true particle classification of the surface.

[Figure]

Line 141: the end of the sentence is not clear for me. When looking at figure2, the wind profile is not significantly modified for z<0.2m. The authors should be more explicit.

**Response**: the fluttering cloth not only enhances the turbulent kinetic energy (Fig. 2) but also modifies the wind profile close to soil surface. We focus on the wind condition close to surface ($z$<0.2 m) because the wind in this region directly drive soil erosion. Although the wind profile is not modified significantly, the change is recognizable. By fitting the wind profile data to Eq. 5, the friction velocity $u_*$ (or shear stress $\rho u_*^2$) could be evaluated. But if $\Psi_m$ is set to 0 for the case of WP, the obtained $u_*$ obviously diverge from the data of Irwin sensor. Only when a non-zero $\Psi_m$ is considered, the deduced $u_*$ agrees with the data of Irwin sensor (Table 2). That is why we state that the wind profile for $z < 0.2$ m modifies to one more similar to wind profile in convective ABL.

We will modify the relevant sentence in the revised version to make it clear.

Table 2: This table needs to be strongly completed and better discussed. I do not retrieve the 25% and 15% difference in $\bar{\tau}$ for fan speeds 7000 rpm and 12000 rpm, respectively, as mentioned line 157, when comparing $u_*$ from the NP and WP profiles.

**Response**: thanks for these suggestions which will be considered in the revised version.

In Line 156-157, the increase of shear stress (not friction velocity) from Irwin sensors is discussed (note that the data of NP is used to calibrate Irwin sensor, i.e. for the case of NP, the shear stresses measured by Irwin sensor and the one deduced from wind profile are same). The relative increase of $\bar{\tau}$ at fan speed 7000 rpm should be $(0.32^2-0.29^2)/0.29^2=22\%$, and $(0.55^2-0.51^2)/0.51^2=16\%$ for the case of 12000 rpm. We apologize for being too rough with our previous estimates, and the exact numbers will be revised.

Are the $u_*$ from Irwin sensors those obtained for the WP conditions. This should be clearer in the table (move WP to be centered).

**Response**: yes, the table will be modified as follow:

Table 2: Friction velocity $\bar{u}_*$ and roughness length $z_0$ estimated for runs with and with no forced perturbation and different wind-tunnel fan speeds. The Irwin sensor is calibrated based on the data of wind profiles under the NP condition.

| Fan Speed (rpm) | NP Profile ($\psi_m = 0$) $\bar{u}_*$ (m s$^{-1}$) | NP Profile ($\psi_m = 0$) $z_0$ (mm) | WP Profile ($\psi_m \neq 0$) $\bar{u}_*$ (m s$^{-1}$) | WP Profile ($\psi_m \neq 0$) $z_0$ (mm) | WP Profile ($\psi_m \neq 0$) $\eta_m$ (mm$^{-1}$) | WP Irwin $\bar{u}_*$ (m s$^{-1}$) |
|---|---|---|---|---|---|---|
| 7000 | 0.29±0.0087 | 0.0133±0.0044 | 0.31±0.0077 | 0.0150±0.0034 | -0.0229 | 0.32±0.0128 |
| 7500 | / | / | 0.34±0.0062 | 0.0153±0.0025 | -0.0213 | 0.34±0.0064 |
| 8000 | 0.34±0.0068 | 0.0146±0.0024 | 0.36±0.0067 | 0.0159±0.0016 | -0.0200 | 0.37±0.0101 |
| 8500 | / | / | 0.39±0.0051 | 0.0167±0.0013 | -0.0188 | 0.39±0.0069 |
| 9000 | 0.38±0.0083 | 0.0159±0.0020 | 0.41±0.0087 | 0.0156±0.0022 | -0.0178 | 0.42±0.0080 |
| 9500 | 0.40±0.0021 | 0.0129±0.0005 | 0.44±0.0038 | 0.0165±0.0007 | -0.0168 | 0.44±0.0031 |
| 10000 | 0.43±0.0106 | 0.0164±0.0024 | 0.46±0.0083 | 0.0166±0.0016 | -0.0160 | 0.47±0.0109 |
| 10500 | 0.47±0.0091 | 0.0207±0.0025 | / | / | / | / |
| 11000 | 0.47±0.0204 | 0.0175±0.0043 | 0.50±0.0067 | 0.0165±0.0010 | -0.0146 | 0.51±0.0077 |
| 12000 | 0.51±0.0180 | 0.0166±0.0035 | 0.55±0.0033 | 0.0162±0.0008 | -0.0133 | 0.55±0.0085 |

Moreover, there are 4 Irwin sensors in the wind tunnel: how do the authors use them? Is the reported data an average of the four sensors? If so, give the standard deviation.

**Response**: in fact, only the data of one probe is completely extracted and analyzed, and the others are used as backup probes. We will remove the other probes illustrated in Figure 1 to avoid misunderstanding.

[Figure]

More generally, give the standard deviations since there are several repetitions of each experiment. Add the emission rate for each experimental condition and the associated standard deviation.

**Response**: thanks for comment. We will add the associated standard deviation in Table 2. For the data of emission rate, relevant standard deviations have been illustrated as the error bars in the figure 4, 5 and 6.

Why does $z_0$ change significantly for NP experiments: it should be constant unless there is an additive saltation roughness but the WP experiments have an almost constant $z_0$ which rules out this assumption. How accurate is the recovery of $u_*$ and $z_0$ from the wind profile? Does this accuracy depend on the regime (i.e. wind speed)?

**Response**: $z_0$ is actually the aerodynamic roughness which is depended not only on the roughness of surface but also on the wind regime. We obtained the $u_*$ and $z_0$ by fitting the measured wind profile data to Eq. (5) (for the case of NP $\psi_m = 0$). For all of the cases, the coefficients of determination are almost 99%. There are two possible reasons for the deviations of $z_0$ at NP_95 and NP_105. The first is the less repeat times for these two cases (3 for NP_95 and 5 for NP_95). The other conjectured reason is a considerable change of weather condition outdoor. The wind tunnel is connected with outside, and obvious weather changes (strong wind or rain) will slightly influence the flow field in the wind tunnel. The results in Table 2 also show that the introduction of parameters $\psi_m$ in the fitting equation is beneficial to obtain a more stable $z_0$.

We will add some explanations for the deviations of $z_0$ in the revised version.

Figure 4 is the key figure of the paper. It shows that the slight increase of $\bar{\tau}$ in quasi-convective conditions is not sufficient alone to explain the measured differences in the emission rates of the four soils. It implies that the perturbations of the shear stress, $\tau'$, are also responsible for a part of the differences in the emission rates. This is perfectly clear.
**Response**: yes it is.

However, the lack of information on the different soils (S1 to S4) complicates the understanding of why the emission rates of the different soils as reported in figure 4 do not follow the order of $\tau_t$ (as shown in table 3). According to equation 12, the only explanation for that is that the different soils have different values of $\bar{\gamma}$ (as suggested by table 3 for NP experiments) but reasons or at least hypotheses allowing to understand this should be given. Furthermore, what could be the possible explanation for such large differences in $\bar{\gamma}$ observed for S1 and S4 but not for S2 and S3 between the two sets of experiments? I can understand that $\bar{\gamma}$ is affected by quasi-convective turbulence but it is difficult to understand why it would affect the four soils so differently.

**Response**: thanks for comment. We will discuss this in the revised version.

In our opinion, $\tau_t$ represents the difficulty level of surface particles for emission, which depends on particle size, arrangement, moisture content and other surface properties. $\bar{\gamma}$ represents the ability of surface particles to respond to winds exceed the threshold, which is determined by the number of erodible particles and the sensitivity of these particles to local turbulence.

It's also important to note that, for surface with mixed-size particle, just like the ones employed in our experiment, all of the particles drove by an active force exceed relevant threshold should emit. We can imagine that particles with the lowest threshold are most prone to move, than the other particles with bigger or smaller size join the movement with the increase of wind. But the distribution of particle is non-uniform which may lead to a various threshold value obtained by regression analysis of emission flux data (Hard-to-move particles may dominate in mass and thus affect the regression value of critical wind).

There are a least three distributions which may influence the final emission flux, including the distribution of shear stress, the distribution of threshold and the distribution of the fraction of released particle. This makes it very difficult to analyze the data.

In order to better answer the reviewers' questions, we re-analyzed the data. First of all, we removed the measurement results of the lowest wind speed (7000 rpm), because the amount of emitted particles corresponding to this condition is very low, which leads to poor stability and greatly interferes with our analysis results. Regression analysis was performed on the remaining data based on the coordinate system of Log10 ($F$) vs $\tau$, with Eq. (13) as the regression equation and $\tau_t$ and $\bar{\gamma}$ as regression parameters.

It is clear that the perturbations of the shear stress have significant influence on emission rate and these changes could be well expressed by Eq. (13) ($R^2$ very close to 1). But the increase of emission rate could be attributed to the increase of $\bar{\gamma}$ and the decrease of $\tau_t$. Based on existing data, we can't clarify this issue better.

**Table:** Shear stress $\tau_t$ and empirical parameter $\gamma$ for test surfaces.

| | WP | | | NP | | | $\dfrac{\overline{\gamma_{WP}}}{\overline{\gamma_{NP}}}$ | $\dfrac{\overline{\tau_{t\_WP}}}{\overline{\tau_{t\_NP}}}$ |
|---|---|---|---|---|---|---|---|---|
| | $\tau_{t\_WP}$ (N·m$^{-2}$) | $\overline{\gamma_{WP}}$ (s$^2$·m$^{-2}$) | $R^2$ | $\tau_{t\_NP}$ (N·m$^{-2}$) | $\overline{\gamma_{NP}}$ (s$^2$·m$^{-2}$) | $R^2$ | | |
| **S1** | 0.17 | 9.96 | 0.92 | 0.08 | 2.10 | 0.97 | 4.74 | 2.13 |
| **S2** | 0.25 | 3282.92 | 0.96 | 0.28 | 4598.63 | 0.99 | 0.71 | 0.89 |
| **S3** | 0.28 | 2388.90 | 0.99 | 0.33 | 4317.70 | 0.99 | 0.55 | 0.85 |
| **S4** | 0.39 | 3130.74 | 0.96 | 0.34 | 502.10 | 0.99 | 6.24 | 1.15 |

For simplicity, we assign the average of $\tau_{t\_WP}$ and $\tau_{t\_NP}$ to $\tau_t$, and re-performed the regression analysis by only considering $\overline{\gamma}$ as regression parameters, meaning that the influence of perturbations of the shear stress is artificially attribute to the change of $\overline{\gamma}$. The results are shown as follow.

[Figure]

**Figure 6.** Estimated entrainment rates with and without forced perturbation. The dots are experimental data (remove 7000 rpm cases) and lines derive from Eq. (12).

**Table:** Evaluated threshold shear stress $\tau_t$ and regression parameter $\overline{\gamma}$ for test surfaces.

| | $\tau_t$ (N·m$^{-2}$) | WP | | NP | | $\dfrac{\overline{\gamma_{WP}}}{\overline{\gamma_{NP}}}$ |
|---|---|---|---|---|---|---|
| | | $\overline{\gamma_{WP}}$ (s$^2$·m$^{-2}$) | $R^2_{WP}$ | $\overline{\gamma_{NP}}$ (s$^2$·m$^{-2}$) | $R^2_{NP}$ | |
| S1 | 0.13 | 7.12 | 0.89 | 3.30 | 0.86 | 1.96 |
| S2 | 0.27 | 4919.84 | 0.95 | 3605.29 | 0.99 | 1.29 |
| S3 | 0.31 | 4675.62 | 0.97 | 2521.74 | 0.98 | 1.97 |
| S4 | 0.37 | 1996.39 | 0.95 | 1713.19 | 0.95 | 1.34 |

In summary, as the threshold $\tau_t$ was first evaluated in our analysis, the factors affecting the emission rate were mainly attributed to $\gamma$ and the effective shear stress $(\tau\text{-}\tau_t)$. We discussed the influence from the forced perturbation to $\gamma$ (Fig. 6 and Tab. 3) and $(\tau\text{-}\tau_t)$ (Fig. 7). The forced perturbation almost double enhances the average $\gamma$(The effect is greater at lower shear stress).There is not much difference among the four soils.

According to the results of emission rate (as shown in Fig. 6), S2 and S3 are more erodible with more big value of $\bar{\gamma}$. S1 is the weakest-eroded surface. Despite start with very low shear stress, the mass of released particle is very limited, that leads to a low $\bar{\gamma}$. For the case of S4, we believe that the emission at low $\tau$ is mainly corresponding to the most-easy-moved particles (with diameter about 70-100 µm) with low $\tau_t$ and low $\gamma$. As the $\tau$ increase, the bigger particles with large mass (high $\gamma$) and $\tau_t$ start to move. Since we analyzed the final aggregate data, all of these effects are mixed together.

The number given in Table 3 are very precise with two significant digits for all parameters but we have no idea of the uncertainties and how are significant the differences in $\bar{\gamma}$.

Response: thanks for comment. We re-analyzed the data and the determination coefficients will be added to judge the uncertainty of regression parameters. There is no special meaning to the significant digits. We just want to keep consistent in the table.

**Table:** Threshold shear stress $\tau_t$ and empirical parameter $\gamma$ for test surfaces.

| | $\tau_t$ (N·m$^{-2}$) | WP | | NP | | $\dfrac{\overline{\gamma}_{\mathrm{WP}}}{\overline{\gamma}_{NP}}$ |
|---|---|---|---|---|---|---|
| | | $\overline{\gamma}_{\mathrm{WP}}$ (s$^2$·m$^{-2}$) | $R^2_{\mathrm{WP}}$ | $\overline{\gamma}_{NP}$ (s$^2$·m$^{-2}$) | $R^2_{\mathrm{NP}}$ | |
| S1 | 0.13 | 7.12 | 0.89 | 3.30 | 0.86 | 1.96 |
| S2 | 0.27 | 4919.84 | 0.95 | 3605.29 | 0.99 | 1.29 |
| S3 | 0.31 | 4675.62 | 0.97 | 2521.74 | 0.98 | 1.97 |
| S4 | 0.37 | 1996.39 | 0.95 | 1713.19 | 0.95 | 1.34 |

Moreover, the quality of the fits to equation 12 is not given while looking at figure 6 (and despite its small size), these fits seem worse for S1 and S4 than for S2 and S3.

**Response**: thanks for comment. The determination coefficients of the fitted lines will be added. That is right, the fits of S1 and S4 are slight worse.

Lines 200 -201: Does this suggest that the impact of convective turbulence should be rather limited on the total dust emission budget?

**Response**: we want to declare that: for the condition with mean shear stress much bigger than the threshold, the surface particle emission is dominantly influenced by the mean effect of convective turbulence; for the condition with mean shear stress close to the threshold, the change of the distribution of surface shear stress caused by convective turbulence significantly influence the process of surface particle emission. Considering that the

approximated-threshold wind conditions may have a high temporal weight in natural conditions, we believe that this effect needs to be considered.

---

## Author Comment (AC3)

**Response to referee #2's interactive comment on the manuscript "Impact of Turbulence on Aeolian Sand and Dust Entrainment: Results from Wind-tunnel Experiment"**

The authors designed a novel technique in wind tunnel to measure the entrainment rate of various particle sizes under different flow conditions. They show that quasi-convective turbulence increases the surface shear stress and hence substantially enhances the entrainment of sand and dust particles. It is a very novel experiment design, and the results are very enlightening.

**Response**: we much appreciate the positive comments from Prof. Cheng.

Comments:

The authors analyzed the power spectrum and PDF of the quasi-convective turbulence, and believed it is similar to convective eddies in atmospheric boundary layers. But from the generation way, the quasi-convective turbulence is more like a kind of coherent structure observed in the atmospheric boundary layer (Liu and Zheng, 2021). Please give some explanation and discussion.

**Response**: thanks for the insight comment and valuable suggestion. The purpose of the arrangement of fluttering cloth in our experiment is to generate different distribution of surface shear stress which is supposed to be significant for surface particle emission. We verified the implementation effect by measuring the distribution of surface shear stress, and analyzed the wind profile and the wind energy spectrum to expose its influence on the boundary layer flow. We indistinctly call it quasi-convective turbulence without deep analysis on eddy structure. Thanks to the reviewer for reminding that our modeled turbulence is similar to the case observed in the field. We will add some explanation and discussion in the revised version.

In line 119-123, how to get $z_{0ABL}$?

**Response:** $Z_{0ABL}$ is set to 3.10 mm, which is suggested to be 2~5mm over concrete, flat desert and tidal flat (Wieinga, 1993). The information is provided in the caption of Figure 2.

In line 141, should "z<0.2m" be "z>0.2m"?

**Response:** actually it is z<0.2m. We focus on the wind condition close to surface (z<0.2 m), because the wind in this region directly drive soil erosion. And the surface shear stress is deduced form the wind profile below 0.2m in the case without fluttering cloth.

Reference

Liu, H. Y. and X. J. Zheng, 2021. Large-scale structures of wall –bounded turbulence in single-and two-phase flows: advancing understanding of the atmospheric surface layer during sandstroms. Flows, 1 E5.

---

## Author Comment (AC4)

**Response to referee #3's interactive comment on the manuscript**

**"Impact of Turbulence on Aeolian Sand and Dust Entrainment:**

**Results from Wind-tunnel Experiment"**

This paper uses wind tunnel simulation and modelling to investigate the response of the sediment entrainment rate to large scale, quasi-convective eddies in boundary-layer air flows. This is a very important topic of interest within the aeolian research community, such that this paper is well aligned with current research needs.

With moderate revision, particularly the inclusion of a great deal of missing information, it may be suitable for publication.

**Response:** *Many thanks for Prof. Cheryl McKenna Neuman for her constructive comments. We will supplement the missing information in the revised version and improve the quality of the manuscript as possible as we can.*

**Minor corrections:**

Title: Define the term 'dust' in the context of this study. You should be clear that you actually do not measure/profile the aerosol concentration of the airflow in this study, so the title is misleading. Perhaps you should say 'silt' instead or drop it altogether.

**Response:** *thanks for suggestion. This work does not specifically deal with dust. In fact, we study the entrainment of particles with different sizes. So we prefer to change the title to:* **"Impact of Turbulence on Aeolian Particle Entrainment: Results from Wind-tunnel Experiment".**

Line 20: Circular argument or poor wording, "turbulent flow is also turbulent"

**Response:** *thanks for pointing it out. We will modify it as "…hence saltation driven by atmospheric boundary-layer (ABL) flows is also turbulent."*

Lines 27-28: Poor wording - "impact" (geophysical process associated with saltation) should be "effect". In addition, "flow is turbulent, and aeolian processes also." Aeolian processes are affected by turbulence but are stochastic.

**Response:** *agree, "effect" will be better. We will modify it in the revision version.*
*The next sentence will be changed to:"In reality, the flow and the flow-driven aeolian processes are both impossible to stabilize. Although people can analyze the relevant average variables, the existence of nonlinear relations (e.g., Eq. 1) makes it difficult to determine the quantitative relations between the average variables."*

Line 74: "scan valve" perhaps should read "scanivalve"

**Response:** *sorry for our mistake. We will revise it as "scanivalve" in the revision.*

Lines 84-84: Define what you mean by 'flutter" versus "flap".

**Response:** *"flutter" corresponds to the delicate swing due to the soft texture of the cloth, while "flap" relates to quick swing determined by the structure of the cloth.*

Line 80: Define what you mean by "positive skewness" with regard to vertical velocity fluctuation. Does positive refer to eddies bursting up away from the bed (+ve w) and negative to eddies sinking down toward the bed (-ve w)? This section is a bit vague.

**Response:** *The "positive skewness" here means the horizontal velocity distribution is not symmetrical as the Gaussian distribution, but towards to positive direction (more weight for the part large than mean value), as shown in Fig. 2b. We will add more description on it in the revision.*

Line 89: The fan speed in rpm is meaningless to the reader. Provide free stream velocities for WP instead.

**Response:** *thanks for the suggestion. We will use the axis wind speed of the incoming flow to replace the fan speed.*

Line 93: Either provide the standard deviations to indicated the sorting for each test sediment or a figure showing the full distribution of particle diameter within each sample. This is important as most test beds will undergo some degree of grain scale armouring when subjected to an air flow and this is likely to be strongly affected by τ'.

**Response:** *thanks for the suggestion. We will add the full distribution of particle diameter within each sample.*

Figures Much too small and almost impossible to read.

**Response:** *thanks for the suggestions. All figures will be improved accordingly.*

**Symbols:**
A number of mathematical symbols appear in the manuscript well before they are defined and sometimes, they are either not defined at all or the symbol has an inconsistent typeface.

**Response:** *thanks for pointing it out. We will check the text carefully and add the lost definition. If necessary, a list of relevant symbols will be added as an appendix.*

Line 165 Specifically you mean the net mass loss (integrated over time).

**Response:** *yes it is the net mass loss. We only weighted the tray after a run of experiment with time of $\Delta T_i$ to determine the loss the mass. The sentence will be modified to make it clear.*

Lines 163 and 170 "Emission rate" interchanged with the "entrainment rate". The former is usually used when talking about dust and the later with the mass transport of sand. This may confuse some readers. Since you are not measuring the dust flux (aerosol concentration) then if might be best to just say "entrainment rate" everywhere.

**Response:** *agree, we will consistently use "entrainment rate".*

**Substantive comments:**
Figure 1 and Section 2. This is the weakest section of the paper with a great deal of missing information and weak justification of the experimental design.

**Response:** *thanks for suggestion, we will improve these parts in revision.*

**Flapping cloth:**

i) This is not a novel solution for creating turbulence in wind tunnels. Provide one or more references to previous work.

**Response:** *We essentially want to produce a flow that leads to a different distribution of surface shear stress. In the experiment, we measured the distribution of surface shear stress and verified the effect of this cloth. We speculated that the effect of cloth is not only to create turbulence, but also to cause a flow effect similar to convection. We didn't focus on the details of flow because they are not the main topic of this article. However, thanks for the reviewer's reminder, we will adjust the expression of this part to make it more appropriate. Previous works of convection simulation (**water tanks**, e.g., Willis and Deardorff, 1974, Yuan et al., 2013; **saline tanks**, e.g., Hibberd and Sawford, 1994; **thermally-stratified wind tunnels**, Hancock and Hayden, 2018) in laboratory will be added as references. In relation to comments iv), it is important to note that for conventional wind-tunnels, to the best of our knowledge, there has been no previous reports of simple ways to generate turbulence similar to convective turbulence which in terms of probability density function has a positive skewness. The positive skewness of convective turbulence plays an important role in aeolian particle entrainment.*

ii) Details of the flapping cloth are missing. What type and weight of fabric? Width is given but more importantly, how long was the sheet? What was the wavelength, amplitude and frequency of its oscillation?

**Response:** *thanks for suggestion. We will add the details of the cloth, including type, weight and size. Since how the cloth changed the flow was not the focus of this experiment, the wavelength, amplitude, frequency of the oscillation were not investigated and recorded in the experiment. But this is a very interesting topic, we can do some special researches in the future work.*

iii) Rationale is not provided for the placement of the cloth relative to the test surface. Is the elevation scaled with the length of the cloth or distance from the leading edge of the sample tray?

**Response:** *As mentioned above, the focus of this paper is not to study the effect of cloth on flow. What we're more interested in at that time is whether this cloth in the wind tunnel can provide the conditions that we need for our experiment, i.e. whether it can produce quasi-convective flow and shear with different distributions on the surface. We did a series of tests before the experiment to check the effect of the cloth, and empirically determine the length of the cloth (1.5m), the height of the arrangement (0.7m) and the distance from the leading edge of the sample tray (1.5m). We will add this information in revision.*
*Of cause, the rationale for the placement of the cloth relative to the test surface in significant and deserves further study.*

iv) Spires are much more commonly used to generate large scale eddies in wind tunnel simulation work and these are typically placed upwind of the roughness element array. There is also a large literature on the effect of spires on turbulent flow and shear. Why did you elect to use a flapping cloth over spires?

**Response:** *it is indeed that several methods have been proposed to generated turbulence in wind tunnels, including spires, roughness element, grid etc. But all of these methods are for neutral boundary layers, and inefficient for the simulation of large eddies commonly observed in convective ABLs. To generate convective turbulence usually requires the use of additional thermal forcing from the surface (**EnFlo stratified flow wind tunnel**, Hancock et al., 2013; Hancock and Farr, 2014; Hancock and Zhang, 2015; Hancock and Hayden, 2018; **controlling temperature of recirculating air and floor panels,** Inagaki et al., 2012; Zhang et al., 2013; Kanda and Yamao, 2016; **thermally-stratified wind tunnels**, Marucci and Hayden, 2018; Marucci and Carpentieri, 2020). To apply surface heating requires normally a large wind tunnel. In any case, to the best of our knowledge, studying the effect of convective eddies on aeolian processes has never been done in wind-tunnel experiments, mainly because we have so far no adequate means to generate convective turbulence in a wind tunnel for aeolian experiments. ,*

*For this work, our hypothesis is that the change of the distribution mode of surface shear caused by large eddies (like convection) may significantly influence aerodynamic entrainment, and this influence could not be determined by the mean surface shear. Therefore, we need to first simulate flow conditions similar to convection to produce distinct distribution of surface shear in wind tunnel, which is the reason for the use of the cloth. Although  we have not studied the intrinsic mechanism how cloth induces large eddies, so far, it does not affect our research on the quantitative characterization of aerodynamic entrainment. This study does provide an inexpensive and practical way to generate the quasi-convective eddies without involing surface heating..*

*More information about the motivation for the using of cloth will be added in the revision.*

**Pitot tubes:**
i) Provide dimensions. Pitot tubes come in a range of sizes, while large tubes places in a fixed vertical array can initiate flow stagnation – bluff body effect.
**Response:** *thanks for comment. There are two types of pitot tubes used in this work. The outer diameter of the small pitot tubes is 1mm, and inner diameter is 0.5mm. 11 these pitot tubes make up a wind profiler, which could simultaneously measure the wind speed profiles. A bigger one with outer diameter of 5mm and inner diameter of 2mm is used to standardize all the small pitot tubes. We will add them in the revision. We will add the size information of Pitot tubes in the revision.*

ii) What did you sample the pressure difference with? State precision and sampling time.
**Response:** *thanks for comment. The sampling frequency is 1 Hz, and the mean horizontal velocity is averaged by 5 minutes. We will add the sampling information of Pitot tubes in the revision.*

**Measurement of the mass transport rate:**
i) One of the greatest challenges with accuracy in measuring mass in the lab using an electronic balance is drift associated with changing air pressure in the room. So it is

possible that any instantaneous fluctuation in F is the a consequence of both the change in mass of particles in the tray and the pressure perturbation in the flow. This would be particularly exacerbated by the presence of large scale eddies. Please explain in detail how you accounted for this in your analysis. Was the total mass loss obtained from measurements in the absence of an airflow or did you average the instantaneous mass transport rate throughout the test?

**Response:** *sorry for this misunderstanding. In fact, we did not weight the tray instantaneously. The tray is only weighted after a test to calculate the average entrainment rate. We will revise the relevant wording to eliminate misunderstanding.*

*The revision will be: "Each tray will be weighted by an electronic balancer with a precision of 0.01 g in the range 5 kg, which measures the total mass loss from the tray after a test."*

Hot wire anemometer and Irwin sensors:
ii) Please state in line 70 that the instrument only samples in 1D
**Response:** *thanks for comment, we will add it in the revision.*

iii) Was the wire ruggedized to withstand particle impact during saltation? If so, how did this affect sensitivity (time constant and precision)?
**Response:** *the wire anemometer was used only in clean-flow to protect from particle impact.*

iv) Please provide exact dimensions of the Irwin sensor rather than a citation. The height of the central port does vary from study to study and is important to know relative to the roughness of the bed surface. Describe also the roughness of the bed surface upwind of the sample trays. Was it smooth or roughened to give a suitable aerodynamic roughness matched to that of the sand in the tray?
**Response:** *the diameter of inner port is 1.65mm, and 1.75mm in height; the diameter of the outer port is 2.57mm; the diameter of the sensor is 12.5mm. The bed surface upwind is cover by a sandpaper (40 mesh) as follow picture. We will add them in the revision.*

**Section 2.1 Results**
**Table 1**
i) Why so many missing experiments? Indeed, there are so few experiments at the 0.5 X1000 rpm increment that it might be better to exclude these data altogether. Add the freestream velocity values for the NP experiments, as rpm will make little sense to the reader.
**Response:** *actually, the cases of 0.5 X1000 rpm are supplementary tests. In the case of large surface shear, the erodible surface (S2 and S3) rapidly appeared obvious surface-concave which would affect the test results. So we added several test in low surface shear. We will explain this in revision.*

*We will use the axis wind speed of the incoming flow to replace the fan speed.*

ii) Describe the bed surface composition for your control runs without a tray/soil.

**Response:** *sorry for the loss of relevant information, we will add the description of the tunnel floor.*

**Figure 3**

i) I am not clear on the surface/sediment texture characteristics pertaining to these data. Have the data for all textures been lumped together?

**Response:** *Figure 3 illustrates the average wind speed profiles over the wind tunnel floor for all of the tested cases. There is only one texture of wind tunnel floor and the tested particles are filled in trays mounted flush to the tunnel floor. The tray is small in area and the surface is smoothed, so we believe that its influence on the wind profile is very limited. We will add descriptions of experimental conditions to make this information clearer.*

ii) The NP vertical profiles of the total velocity look great – a deep well-structured boundary-layer flow following the law of the wall. The WP profiles also look very good, but I think even more might be inferred from them than what is provided. There is indeed a high degree of flow stagnation in the upper part of the profile above 20 cm associated with partitioning of momentum to the flapping cloth turbulent but also turbulent energy dissipation within the wake flow downwind. However, little is said about the proportionate acceleration of the air flow (~0.5 m/s) at all levels below 20 cm in all WP experiments. This is unexpected and could be a wind tunnel artifact associated with a small degree of compression of the flow that was redirected beneath the cloth and between the confining walls.

**Response:** *Yes, for the cases of NP, the wind profiles are normal. According to the analysis of regression based on the logarithmic law (Eq. 5 with $\Psi_m=0$), we obtained the friction velocity for each case to calibrate the Irwin sensor.*

*When we use the same method to analyze the data of WP conditions, there are a great difference between the estimated friction wind speed and ones measured by Irwin probe. However, if we take the effect of $\psi_M$ into account in the regression equation (i.e., not set to zero), the obtained friction is very close to Irwin's result. Therefore, we speculate that the cloth induce a flow similar to the convection in ABL.*

*We noticed an increase in wind speed near the surface, which we think is a result of the presence of the cloth increasing the exchange of horizontal momentum in the vertical direction.*

*Unfortunately, we have not measured more wind field information besides the wind speed profile (Pitot tube) and the time series of the wind speed at a certain height (1-dimensional hot wire), so there is impossible to do a deeper analysis of the turbulence structure induced by the cloth.*

iii) It is really unfortunate in this study that the vertical component of the total wind speed was not isolated and sampled, as required for analysis of the eddy structure. As first identified by Thom (1975) for truly unstable conditions with convection, mechanical effects (roughness) dominate the near surface flow, but in moving away from the surface the eddies are stretched vertically and the momentum flux is enhanced. Since the vertical velocity is generally one to two orders of magnitude smaller than the

horizontal component, the total wind speed as sampled in this study cannot be particularly sensitive to the eddy perturbation that was intentionally created.

**Response:** *thanks for comment. The simulation and analysis of convective flow in wind tunnel is indeed an interesting and challenging topic. We will continue to focus on this in the future.*

**Table 2**

i) Again, I am not clear on whether the data have been averaged across all sediment textures or this is just one example for a specific texture.

**Response:** *as we responded above, there is only a smoothed and mounted sediment surface with a small area employed to test. The wind field is mainly affected by the wind tunnel floor which is unchanged during the experiment.*

ii) Why is the threshold friction velocity also not reported here?

**Response:** *Table 2 only provides the parameters of wind filed. The threshold friction velocity is a property parameter of the grain surface, which is provided in Table 3.*

iii) Over what elevations were the wind speed data used to calculate u*? If the boundary layer depth is taken to be 20 cm for the WP experiments and the inner constant stress region 15-20% of that, then your calculation might only be based on 4 data points. That is, the shear stress values for the WP experiments would carry greater error than those for the NP experiments. How does this uncertainty compare with the roughly 7% increase in u* associated with introduction of the flapping cloth?

**Response:** *thanks for comment. There is indeed uncertainty in estimating friction velocity from wind profile. In fact, the shear stresses or friction velocities in the Figs. 4-7 are provided by Irwin probe data.*

iv) Under what conditions were the Irwin sensors calibrated – NP or WP? This has a large bearing on an explanation for why the Irwin sensor u* values sit slightly higher than for the WP data.

**Response:** *the Irwin sensor is calibrated under NP condition which has been validated for well estimating the friction velocity from wind profile.*

**Section 2.1.3**

Line 171

"Convective turbulence is much more efficient in lifting particles into the air". Caution should be exercised in making such a broad statement. This may well be true for very light particles (aerosols) entering suspension, given a settling velocity that is lower than the vertical velocity at which eddies burst away from the bed surface, but not so for very large sand particles. Once again, it is unfortunate that you can only infer such effects because you don't have the vertical component measurements.

**Response:** *thanks for comment. Our results on four tested surfaces with different average particle sizes show that convection facilitates particle entrainment. We prove that this is not only caused by the increase of mean surface shear stress, but also by the change of the distribution surface shear stress.*

*Anyway, we're going to use more rigorous language in revision.*

Figures 4 and 5

There is no question that positive skewness in the instantaneous bed shear stress should increase the particle entrainment rate at low values near threshold, but the fact that the near surface flow is accelerated (especially at lower elevations) by insertion of the flapping cloth is likely to be equally if not more important in this particular study.

**Response:** *thanks for comment. We believe that the increase of wind speed in the near-surface region is reflected in the increase of surface average shear stress. But the increase of average shear cannot completely correspond to the increase of particle entrainment (Fig. 4). Only by taking the influence of the change of shear stress distribution into account, the measured results of entrainment are well explained.*

**Conclusions**

Logic dictates and your study shows that intermittent generation of large shear stresses on the bed surface enhances the entrainment of sand and silt sized particles. This should be qualified by some form of statement to indicate that this effect is greatest at low wind speeds around threshold and when transport is intermittent. In really strong winds mechanical or impact entrainment dominates, while the particle borne stress largely determines the transport under saturated conditions.

**Response:** *thanks for comment. According to Fig 7, it obviously shows that the effect of the distribution of surface shear is greatest at low wind speeds around threshold and when transport is intermittent. We will modify relevant conclusion according to the suggestion of reviewer.*

"we showed that wind-tunnel turbulence lacks energy containing eddies even compared with ABL flows in neutral conditions, highlighting the deficiency of traditional wind-tunnel experiments on aeolian studies". Again, this is a bit of an overstatement. Most traditional wind tunnel studies have modelled saturated flows (see above) where the particle cloud itself is so dense that it alters the turbulence structure. Since the development of fast response pressure transmitters, cross-wire probes, and laser Doppler anemometers more than 2 decades ago, aeolian researchers working in wind tunnels have indeed provided detailed measurements of the turbulence structure and intensity associated with sediment entrainment and transport while also investigating the effects of wind gusting (e.g. Li and McKenna Neuman 2012, 2014 etc). Similarly, there are many means by which we can and do generate large scale eddies in wind tunnel simulation and these are widely practiced in engineering applications, particularly in investigations of wind loading on urban structures.

**Response:** *thanks a lot for the comment and providing information. We will modify relevant part to make it more rigorous.*

---

## Author Comment (AC5)

Wind profiler

Expected wind direction

Reference tube

**Irwin senor dimensions**
1.75 mm (height if inner port above surface)
1.65 mm (dia. of inner port)
2.57 mm (dia. of outer port)

13.5mm

**Tray dimension**
150 mm (length)
285 mm (width)
13 mm (depth)

Length

Width

---

## Author Response (AR1)

Dear Editor:

Thank you for your kind letter of "acp-2022-76 (author) - file upload for peer-review completion" on April 29, 2022. We have revised the manuscript "Impact of Turbulence on Aeolian Sand and Dust Entrainment: Results from Wind-tunnel Experiment" in accordance with the reviewers' comments, and carefully proof-read the manuscript to minimize typographical, grammatical, and bibliographical errors during last one and half months. Here below is our description on revision according to the reviewers' comments. The font color of the comments is black, the responses is blue, and the revisions is red.

**Response to reviewer #1**

**Comment 1:** This paper aims to investigate the effects of atmospheric turbulence on sediment emission rates. This topic is a central issue in recent investigations by the community with various papers suggesting important effects of turbulence on dust fluxes and their size distribution.

This paper contains two major elements: it implements a simple technique to generate quasi-convective turbulence in a wind tunnel. This is an important advance because the lack of wind tunnel experiments on sediment movement has so far been carried out under neutral or near-neutral conditions which do not allow the study of the role of large eddies on wind erosion. The second major point, well-illustrated by Figure 4, is that the results show that the sediment emission rates are higher during quasi-convective turbulence situations. These are significant contributions that deserve a publication of this paper.

Nevertheless, in its form, some parts need to be better detailed to improve understanding or to avoid unanswered questions. Most figures need to be enlarged.

**Response:** we are most grateful to Prof. Gilles Bergametti for his encouraging comments and constructive suggestions. The Referee pointed out the essential contributions of this work to elucidate the effects of atmospheric turbulence on sediment emission. We have modified all the parts according to the comments and all the figures are enlarged.

**Comment 2:** Lines 17-18: I agree that "it has been shown in numerous studies that equation 1 is valid in general" but some papers have also challenged its validity (e.g., Martin & Kok, Science Advances, 2017; Andreotti, J. Fluid Mech., 2004). It is not the purpose of this paper to discuss this but the wording used could suggest that this debate does not exist.

**Response:** thanks for the comment. We have modified this sentence in the revision (highlight version) in line 17-18:

"Although challenged by some papers (e.g., Martin & Kok, 2017), it has been shown in numerous studies that Eq. (1) is valid in general…".

However, it should be note that the argument of the linear-relationship between saltation flux $Q$ and shear stress $\tau$ (or $u_*^2$) is confined to saturated aeolian flow condition (or steady-state saltation condition) with splash-dominated particle entrainment. The majority of the existing measurements do not support the $Q \sim \tau$ linear relationship. In our study, aerodynamic entrainment is dominant, and Eq. 1 should be reasonable to be used.

**Comment 3:** Lines 74-75: I understand the first part of the sentence, which seems sufficient in itself. What more do you mean by "by comparing the shear stresses measured by the two devices"?

**Response:** indeed, the latter part of this sentence is redundant. We have deleted it, as shown in the revision (highlight version) in line 100.

**Comment 4:** Lines 92-94: The description of the soils is really limited. On reading, one gets the impression that only the average particle size differs and this is not sufficient to then understand why $\bar{\gamma}$ are so different in table 3.

**Response:** sorry for this negligence on the description of soil. We have measured the size distribution of the employed four soils by using a Microtrac S3500 Laser Diffractometer (Microtrac, Montgomeryville, PA, USA) at the preparing section. We added this information in the revision (highlight version) in line 119-121:

"Four different soils are used in the experiment, labelled as S1-S4. The mean particle sizes of the four soils are respectively 75, 140, 215 and 398 μm. The particle size distributions are approximately log-normal, as shown in Fig.1d, measured by a Microtrac S3500 Laser Diffractometer (Microtrac, Montgomeryville, USA)."

**Comment 5:** Line 141: the end of the sentence is not clear for me. When looking at figure2, the wind profile is not significantly modified for z<0.2m. The authors should be more explicit.

**Response**: the fluttering cloth not only enhances the turbulent kinetic energy (Fig. 2) but also modifies the wind profile close to soil surface. We focus on the wind condition close to surface ($z$<0.2 m) because the wind in this region directly drive soil erosion. Although the wind profile is not modified significantly, the change is recognizable. By fitting the wind profile data to Eq. 5, the friction velocity $u_*$ (or shear stress $\tau$) could be evaluated. But if $\Psi_m$ is set to 0 for the case of WP, the obtained $u_*$ obviously diverge from the data of Irwin sensor. Only when a non-zero $\Psi_m$ is considered, the deduced $u_*$ agrees with the data of Irwin sensor (as shown in Table 2). That is why we state that the wind profile for $z < 0.2$ m modifies to one more similar to wind profile in convective ABL. We modified it in the revision (highlight version) in line 179-183:

"For z < 0.2 m in WP runs, the air flow seems to be accelerated (~ 0.5 m s$^{-1}$). It could be a wind tunnel artifact associated with a small degree of compression of the flow that was redirected beneath the cloth and between the confining walls. For a given fan speed, the fluttering cloth not only enhances the turbulent kinetic energy (Fig. 2) but also modifies the wind profile for z < 0.2 m to one more similar to wind profile in convective ABL, which will be proved by the following analysis."

and in line 190-196:

"The shear stress $\bar{\tau} = \rho \bar{u}_*^2$ (here, air density $\rho$=1.2 kg·m$^{-3}$) is then used to calibrate the shear stress measured using the Irwin sensor, $\bar{\tau}_{\text{Irwin}}$. But if $\psi_m$ is set to 0 for the case of WP, the obtained u$_*$ obviously diverge from the data of Irwin sensor. Only when a non-zero $\psi_m$ is considered, the deduced u$_*$ agrees with the data of Irwin sensor (Table 2). That is why we state that the wind profile for z < 0.2 m modifies to one more similar to wind profile in convective ABL."

**Comment 6:** Table 2: This table needs to be strongly completed and better discussed. I do not

retrieve the 25% and 15% difference in $\bar{\tau}$ for fan speeds 7000 rpm and 12000 rpm, respectively, as mentioned line 157, when comparing $u*$ from the NP and WP profiles. Are the $u*$ from Irwin sensors those obtained for the WP conditions. This should be clearer in the table (move WP to be centered).

**Response**: thanks for these suggestions. We have reorganized Table 2, and marked it more clearer of those $u_*$ either from profile fitting or Irwin sensors for the WP conditions, shown as following:

Table 2: Friction velocity $\bar{u}_*$ and roughness length $z_0$ estimated for runs with and with no forced perturbation and different wind-tunnel fan speeds ($R^2 = 99\%$). The Irwin sensor is calibrated based on the data of wind profiles under the NP condition.

| Fan Speed (rpm) | $U_{501mm}$ (m/s) | NP | | WP | | | |
| | | Profile ($\psi_m = 0$) | | Profile ($\psi_m \neq 0$) | | | Irwin |
| | | $\bar{u}_*$ (m s$^{-1}$) | $z_0$ (mm) | $\bar{u}_*$ (m s$^{-1}$) | $z_0$ (mm) | $\eta_m$ (mm$^{-1}$) | $\bar{u}_*$ (m s$^{-1}$) |
|---|---|---|---|---|---|---|---|
| 7000 | 7.68 | 0.29±0.0087 | 0.0133±0.0044 | 0.31±0.0077 | 0.0150±0.0034 | -0.0229 | 0.32±0.0128 |
| 7500 | 8.32 | / | / | 0.34±0.0062 | 0.0153±0.0025 | -0.0213 | 0.34±0.0064 |
| 8000 | 8.95 | 0.34±0.0068 | 0.0146±0.0024 | 0.36±0.0067 | 0.0159±0.0016 | -0.0200 | 0.37±0.0101 |
| 8500 | 9.56 | / | / | 0.39±0.0051 | 0.0167±0.0013 | -0.0188 | 0.39±0.0069 |
| 9000 | 10.18 | 0.38±0.0083 | 0.0159±0.0020 | 0.41±0.0087 | 0.0156±0.0022 | -0.0178 | 0.42±0.0080 |
| 9500 | 10.80 | / | / | 0.44±0.0038 | 0.0165±0.0007 | -0.0168 | 0.44±0.0031 |
| 10000 | 11.42 | 0.43±0.0106 | 0.0164±0.0024 | 0.46±0.0083 | 0.0166±0.0016 | -0.0160 | 0.47±0.0109 |
| 11000 | 12.55 | 0.47±0.0204 | 0.0175±0.0043 | 0.50±0.0067 | 0.0165±0.0010 | -0.0146 | 0.51±0.0077 |
| 12000 | 13.72 | 0.51±0.0180 | 0.0166±0.0035 | 0.55±0.0033 | 0.0162±0.0008 | -0.0133 | 0.55±0.0085 |

In Line 156-157, the increase of shear stress (not friction velocity) from Irwin sensors is discussed (note that the data of NP is used to calibrate Irwin sensor, i.e., for the case of NP, the shear stresses measured by Irwin sensor and the one deduced from wind profile are same). The relative increase of $\bar{\tau}$ at fan speed 7000 rpm should be $(0.32^2-0.29^2)/0.29^2=22\%$, and $(0.55^2-0.51^2)/0.51^2=16\%$ for the case of 12000 rpm. We apologize for being too rough with our previous estimates. We have modified it in the revision (highlight version) in line 201-203:

"Table 2 shows that forced perturbation leads to an increased $\bar{u}_*$, corresponding to an increase of $\bar{\tau}$ by about 22% at fan speed 7000 rpm and about 16% at fan speed 12000 rpm."

**Comment 7:** Moreover, there are 4 Irwin sensors in the wind tunnel: how do the authors use them? Is the reported data an average of the four sensors? If so, give the standard deviation.

**Response**: in fact, only the data of one probe is completely extracted and analyzed, and the others are used as backup probes. We have removed the other probes illustrated in Figure 1 to avoid misunderstanding, and modified the text in the revision (highlight version) in line 93-94: "An Irwin sensor is mounted on the central axis of the wind tunnel floor and locates upwind of the tray."

**Comment 8:** More generally, give the standard deviations since there are several repetitions of each experiment. Add the emission rate for each experimental condition and the associated

standard deviation.

**Response**: thanks for comment. We have added the associated standard deviation in Table 2, as shown in response of comment 6. For the data of emission rate, relevant standard deviations have been illustrated as the error bars in the figure 4, 5 and 6.

**Comment 9:** Why does $z_0$ change significantly for NP experiments: it should be constant unless there is an additive saltation roughness but the WP experiments have an almost constant $z_0$ which rules out this assumption. How accurate is the recovery of $u_*$ and $z_0$ from the wind profile? Does this accuracy depend on the regime (i.e., wind speed)?

**Response**: $z_0$ is actually the aerodynamic roughness which is depended not only on the roughness of surface but also on the wind regime. We obtained the $u_*$ and $z_0$ by fitting the measured wind profile data to Eq. (5) (for the case of NP $\psi_m = 0$). For all cases, the coefficients of determination are almost 99%. There are two possible reasons for the deviations of $z_0$ at NP_95 and NP_105. The first is the less repeat times for these two cases (3 for NP_95 and 5 for NP_105). The other conjectured reason is a considerable change of weather condition outdoor. The wind tunnel is connected with outside, and obvious weather changes (strong wind or rain) will slightly influence the flow field in the wind tunnel. The results in Table 2 also show that the introduction of parameters $\psi_m$ in the fitting equation is beneficial to obtain a more stable $z_0$. To avoid misunderstanding, we deleted the NP_95 and NP_105 in Table 2.

**Comment 10:** Figure 4 is the key figure of the paper. It shows that the slight increase of $\bar{\tau}$ in quasi-convective conditions is not sufficient alone to explain the measured differences in the emission rates of the four soils. It implies that the perturbations of the shear stress, $\tau'$, are also responsible for a part of the differences in the emission rates. This is perfectly clear. However, the lack of information on the different soils (S1 to S4) complicates the understanding of why the emission rates of the different soils as reported in figure 4 do not follow the order of $\tau_t$ (as shown in table 3). According to equation 12, the only explanation for that is that the different soils have different values of $\bar{\gamma}$ (as suggested by table 3 for NP experiments) but reasons or at least hypotheses allowing to understand this should be given. Furthermore, what could be the possible explanation for such large differences in $\bar{\gamma}$ observed for S1 and S4 but not for S2 and S3 between the two sets of experiments? I can understand that $\bar{\gamma}$ is affected by quasi-convective turbulence but it is difficult to understand why it would affect the four soils so differently.

**Response**: thanks for comment. In our opinion, $\tau_t$ represents the difficulty level of surface particles for emission, which depends on particle size, arrangement, moisture content and other surface properties. $\bar{\gamma}$ represents the ability of surface particles to respond to winds exceed the threshold, which is determined by the number of erodible particles and the sensitivity of these particles to local turbulence.

It's also important to note that, for surface with mixed-size particle, just like the ones employed in our experiment, all the particles drove by an active force exceed relevant threshold should emit. We can imagine that particles with the lowest threshold are most prone to move, then the other particles with bigger or smaller size join the movement with the increase of wind. But the distribution of particle is non-uniform which may lead to a various threshold value obtained by regression analysis of emission flux data (Hard-to-move particles may dominate in mass and thus affect the regression value of critical wind).

There are a least three distributions which may influence the final emission flux, including the distribution of shear stress, the distribution of threshold and the distribution of the fraction of released particle. This makes it very difficult to analyze the data.

In order to better answer the reviewers' questions, we re-analyzed the data. First of all, we removed the measurement results of the lowest wind speed (7000 rpm), because the amount of emitted particles corresponding to this condition is very low, which leads to poor stability and greatly interferes with our analysis results. Regression analysis was performed on the remaining data based on the coordinate system of Log10 ($F$) vs $\tau$, with Eq. (13) as the regression equation and $\tau_t$ and $\bar{\gamma}$ as regression parameters.

It is clear that the perturbations of the shear stress have significant influence on emission rate and these changes could be well expressed by Eq. (13) ($R^2$ very close to 1). But the increase of emission rate could be attributed to the increase of $\bar{\gamma}$ and the decrease of $\tau_t$. Based on existing data, we can't clarify this issue better.

**Table:** Shear stress $\tau_t$ and empirical parameter $\gamma$ for test surfaces.

| | WP | | | NP | | | $\dfrac{\gamma_{WP}}{\gamma_{NP}}$ | $\dfrac{\overline{\tau_{t\_WP}}}{\overline{\tau_{t\_NP}}}$ |
|---|---|---|---|---|---|---|---|---|
| | $\tau_{t\_WP}$ (N·m$^{-2}$) | $\overline{\gamma_{WP}}$ (s$^2$·m$^{-2}$) | $R^2$ | $\tau_{t\_NP}$ (N·m$^{-2}$) | $\overline{\gamma_{NP}}$ (s$^2$·m$^{-2}$) | $R^2$ | | |
| **S1** | 0.17 | 9.96 | 0.92 | 0.08 | 2.10 | 0.97 | 4.74 | 2.13 |
| **S2** | 0.25 | 3282.92 | 0.96 | 0.28 | 4598.63 | 0.99 | 0.71 | 0.89 |
| **S3** | 0.28 | 2388.90 | 0.99 | 0.33 | 4317.70 | 0.99 | 0.55 | 0.85 |
| **S4** | 0.39 | 3130.74 | 0.96 | 0.34 | 502.10 | 0.99 | 6.24 | 1.15 |

For simplicity, we assign the average of $\tau_{t\_WP}$ and $\tau_{t\_NP}$ to $\tau_t$, and re-performed the regression analysis by only considering $\bar{\gamma}$ as regression parameters, meaning that the influence of perturbations of the shear stress is artificially attribute to the change of $\bar{\gamma}$. The results are shown as follow.

[Figure]

**Figure 6.** Estimated entrainment rates with and without forced perturbation. The dots are experimental data (remove 7000 rpm cases) and lines derive from Eq. (12).

**Table:** Evaluated threshold shear stress $\tau_t$ and regression parameter $\bar{\gamma}$ for test surfaces.

| | $\tau_t$ (N·m$^{-2}$) | WP | | NP | | $\dfrac{\overline{\gamma_{WP}}}{\overline{\gamma_{NP}}}$ |
|---|---|---|---|---|---|---|
| | | $\overline{\gamma_{WP}}$ (s$^2$·m$^{-2}$) | $R^2_{WP}$ | $\overline{\gamma_{NP}}$ (s$^2$·m$^{-2}$) | $R^2_{NP}$ | |
| S1 | 0.13 | 7.12 | 0.89 | 3.30 | 0.86 | 1.96 |
| S2 | 0.27 | 4919.84 | 0.95 | 3605.29 | 0.99 | 1.29 |
| S3 | 0.31 | 4675.62 | 0.97 | 2521.74 | 0.98 | 1.97 |
| S4 | 0.37 | 1996.39 | 0.95 | 1713.19 | 0.95 | 1.34 |

In summary, as the threshold $\tau_t$ was first evaluated in our analysis, the factors affecting the emission rate were mainly attributed to $\gamma$ and the effective shear stress ($\tau$-$\tau_t$). We discussed the influence from the forced perturbation to $\gamma$ (Fig. 6 and Table 3) and ($\tau$-$\tau_t$) (Fig. 7). The forced perturbation almost double enhances the average $\gamma$(The effect is greater at lower shear stress). There is not much difference among the four soils.

According to the results of emission rate (as shown in Fig. 6), S2 and S3 are more erodible with more big value of $\bar{\gamma}$. S1 is the weakest-eroded surface. Despite start with very low shear stress, the mass of released particle is very limited, that leads to a low $\bar{\gamma}$. For the case of S4, we believe that the emission at low $\tau$ is mainly corresponding to the most-easy-moved particles (with diameter about 70-100 μm) with low $\tau_t$ and low $\gamma$. As the $\tau$ increase, the bigger particles with large mass (high $\gamma$) and $\tau_t$ start to move. Since we analyzed the final aggregate data, all of these effects are mixed together.

However, this is not the main point of our works. To avoid dispersing the main idea of the article, we modified the text in the revision (highlight version) in line 246-249:
"The low value of $\bar{\gamma}$ for S1 is supposed to be caused by the wide distributed particle size, leading to an increased mean threshold friction velocity when wind speed increases, which means constant threshold is not suitable in this situation. However, this is beyond the scope of this work, which mainly focus on the comparison of the WP and NP conditions."

**Comment 11:** The number given in Table 3 are very precise with two significant digits for all parameters but we have no idea of the uncertainties and how are significant the differences in $\bar{\gamma}$.

**Response:** thanks for comment. We have re-analyzed the data and the determination coefficients to judge the uncertainty of regression parameters. There is no special meaning to the significant digits. We just want to keep consistent in the Table 3. The modification is shown as:

**Table 3:** Threshold shear stress $\tau_t$ and empirical parameter $\gamma$ for test surfaces.

| Soil type | $\tau_t$ (N·m$^{-2}$) | WP | | NP | | $\dfrac{\overline{\gamma_{WP}}}{\overline{\gamma_{NP}}}$ |
|---|---|---|---|---|---|---|
| | | $\overline{\gamma_{WP}}$ (s$^2$·m$^{-2}$) | $R^2_{WP}$ | $\overline{\gamma_{NP}}$ (s$^2$·m$^{-2}$) | $R^2_{NP}$ | |
| S1 | 0.13 | 7.12 | 0.89 | 3.30 | 0.86 | 1.96 |
| S2 | 0.27 | 4919.84 | 0.95 | 3605.29 | 0.99 | 1.29 |
| S3 | 0.31 | 4675.62 | 0.97 | 2521.74 | 0.98 | 1.97 |
| S4 | 0.37 | 1996.39 | 0.95 | 1713.19 | 0.95 | 1.34 |

**Comment 12:** Moreover, the quality of the fits to equation 12 is not given while looking at figure 6 (and despite its small size), these fits seem worse for S1 and S4 than for S2 and S3.

**Response**: thanks for comment. The determination coefficients of the fitted lines is added in Table 3. That is right, the fits of S1 and S4 are slight worse.

**Comment 13:** Lines 200 -201: Does this suggest that the impact of convective turbulence should be rather limited on the total dust emission budget?

**Response**: we want to declare that: for the condition with mean shear stress much bigger than the threshold, the surface particle emission is dominantly influenced by the mean effect of convective turbulence; for the condition with mean shear stress close to the threshold, the change of the distribution of surface shear stress caused by convective turbulence significantly influence the process of surface particle emission. Considering that the approximated-threshold wind conditions may have a high temporal weight in natural conditions, we believe that this effect needs to be considered.

**Response to reviewer #2**

**Comment 1:** The authors designed a novel technique in wind tunnel to measure the entrainment rate of various particle sizes under different flow conditions. They show that quasi-convective turbulence increases the surface shear stress and hence substantially enhances the entrainment of sand and dust particles. It is a very novel experiment design, and the results are very enlightening.

**Response**: we much appreciate the positive comments from Prof. Cheng.

**Comment 2:** The authors analyzed the power spectrum and PDF of the quasi-convective turbulence, and believed it is similar to convective eddies in atmospheric boundary layers. But from the generation way, the quasi-convective turbulence is more like a kind of coherent structure observed in the atmospheric boundary layer (Liu and Zheng, 2021). Please give some explanation and discussion.

**Response**: thanks for the insight comment and valuable suggestion. The purpose of the arrangement of fluttering cloth in our experiment is to generate different distribution of surface shear stress which is supposed to be significant for surface particle emission. We verified the implementation effect by measuring the distribution of surface shear stress and analyzed the wind profile and the wind energy spectrum to expose its influence on the boundary layer flow. We indistinctly call it quasi-convective turbulence without deep analysis on eddy structure. Thanks to the reviewer for reminding that our modeled turbulence is similar to the case observed in the field. We added some explanation and discussion in the revision (highlight version) in line 133-137:

"It is shown that the effect of cloth not only enhances the average value of instantaneous wind speed, but also causes the probability of strong wind to increase in the distribution of instantaneous wind speed. While depending on the fluttering mode of the cloth, the quasiconvective turbulence has coherent structures as observed in convective ABLs (Liu and Zheng, 2021). As our main interest is how convective eddies affects aeolian particle entrainment, we did not study the intrinsic mechanisms for how cloth induces large eddies."

**Comment 3:** In line 119-123, how to get $z_{0ABL}$?

**Response:** $Z_{0ABL}$ is set to 3.10 mm, which is suggested to be 2~5mm over concrete, flat desert and tidal flat (Wieinga, 1993). The information is provided in the caption of Fig. 2.

**Comment 4:** In line 141, should "z<0.2m" be "z>0.2m"?

**Response:** actually, it is z<0.2m. We focus on the wind condition close to surface (z<0.2 m) because the wind in this region directly drive soil erosion. And the surface shear stress is deduced form the wind profile below 0.2m in the case without fluttering cloth.

**Response to reviewer #3**
**Comment 1:** This paper uses wind tunnel simulation and modelling to investigate the response of the sediment entrainment rate to large scale, quasi-convective eddies in boundary-layer air flows. This is a very important topic of interest within the aeolian research community, such that this paper is well aligned with current research needs.
With moderate revision, particularly the inclusion of a great deal of missing information, it may be suitable for publication.
**Response:** Many thanks to Prof. Cheryl McKenna Neuman for her constructive comments. We have supplemented the missing information in the revised version and improve the quality of the manuscript as possible as we can.

**Comment 2:** Title: Define the term 'dust' in the context of this study. You should be clear that you actually do not measure/profile the aerosol concentration of the airflow in this study, so the title is misleading. Perhaps you should say 'silt' instead or drop it altogether.
**Response:** Thanks for the suggestion. This work does not specifically deal with dust. In fact, we study the entrainment of particles of different sizes. So, we prefer to change the title to: "Impact of Turbulence on Aeolian Particle Entrainment: Results from Wind-tunnel Experiment".

**Comment 3:** Line 20: Circular argument or poor wording, "turbulent flow is also turbulent"
**Response:** Thanks for pointing it out. We have modified it in the revision (highlight version) in line 20-21:
"…hence saltation driven by atmospheric boundary-layer (ABL) flows is also turbulent."

**Comment 4:** Lines 27-28: Poor wording - "impact" (geophysical process associated with saltation) should be "effect". In addition, "flow is turbulent, and aeolian processes also." Aeolian processes are affected by turbulence but are stochastic.
**Response:** We agree with your opinion that the word "effect" will be better. We have modified it in the revision (highlight version) in line 28-31:
"In reality, the flow and the flow-driven aeolian processes are both impossible to stabilize.

Although people can analyze the relevant average variables, the existence of nonlinear relations (e.g., Eq. 1) makes it difficult to determine the quantitative relations between the average variables."

**Comment 5:** Line 74: "scan valve" perhaps should read "scanivalve"
**Response:** sorry for our mistake. We have revised it as "scanivalve" in the revision.

**Comment 6:** Lines 84-84: Define what you mean by 'flutter" versus "flap".
**Response:** "flutter" corresponds to the delicate swing due to the soft texture of the cloth, while "flap" relates to quick swing determined by the structure of the cloth.

**Comment 7:** Line 80: Define what you mean by "positive skewness" with regard to vertical velocity fluctuation. Does positive refer to eddies bursting up away from the bed (+ve w) and negative to eddies sinking down toward the bed (-ve w)? This section is a bit vague.
**Response:** The "positive skewness" here means the horizontal velocity distribution is not symmetrical as the Gaussian distribution, but towards to positive direction (more weight for the part large than mean value), as shown in Fig. 2b. We added more description to it in the revision (highlight version) in line 103-104:
"While horizontal velocity fluctuations are approximately Gaussian distributed, vertical velocity fluctuations are typically non-Gaussian with a positive skewness, resulting in a positive skewed probability distribution of surface shear stress."

**Comment 8:** Line 89: The fan speed in rpm is meaningless to the reader. Provide free stream velocities for WP instead.
**Response:** Thanks for the suggestion. We modified it in the revision (highlight version) in line 114-116:
"For our experiment, the fan speed is fixed for each run between 7000 and 12000 rpm with interval of 1000 rpm, and the corresponding inlet free wind speed is between 7.7 m s$^{-1}$ and 13.7 m s$^{-1}$."
Also added incoming free speed $U_{501mm}$ in Table 2.

**Comment 9:** Line 93: Either provide the standard deviations to indicated the sorting for each test sediment or a figure showing the full distribution of particle diameter within each sample. This is important as most test beds will undergo some degree of grain scale armouring when subjected to an air flow and this is likely to be strongly affected by τ'.
**Response:** Thanks for the suggestion. We added the full distribution of particle diameter within each sample, shown in Fig.1d. The text has been modified in the revision (highlight version) in line 119-121:
"The mean particle sizes of the four soils are respectively 75, 140, 215 and 398 μm. The particle size distributions are approximately log-normal, as shown in Fig.1d, measured by a Microtrac S3500 Laser Diffractometer (Microtrac, Montgomeryville, USA)."

**Comment 10:** Figures Much too small and almost impossible to read.
**Response:** Thanks for the suggestions. All figures have been enlarged.

**Comment 11:** A number of mathematical symbols appear in the manuscript well before they are defined and sometimes, they are either not defined at all or the symbol has an inconsistent typeface.

**Response:** Thanks for pointing it out. We have checked the text carefully and added the lost definition.

**Comment 12:** Line 165 Specifically you mean the net mass loss (integrated over time).

**Response:** Yes, it is the net mass loss. We only weighted the tray after a run of experiments with the time of $\Delta T_i$ to determine the loss of the mass. The sentence has been modified in the revision (highlight version) in line 83-84:

"Each tray is weighted before and after each test by an electronic balancer with a precision of 0.01 g in the range 5 kg, to determine the net mass loss of the tested surface."

and line 213:

"where $\Delta m_i$ is the net mass loss (integrated over $\Delta T_i$) from the tray during in the i$^{th}$ run with runtime $\Delta T_i$…"

**Comment 13:** Lines 163 and 170 "Emission rate" interchanged with the "entrainment rate". The former is usually used when talking about dust and the later with the mass transport of sand. This may confuse some readers. Since you are not measuring the dust flux (aerosol concentration) then if might be best to just say "entrainment rate" everywhere.

**Response:** Agree, we have consistently used "entrainment rate" in the revision.

**Comment 14:** Figure 1 and Section 2. This is the weakest section of the paper with a great deal of missing information and weak justification of the experimental design.

Response: Thanks for suggestion, we have improved these parts in revision.

**Comment 15:**

**Flapping cloth:**

i) This is not a novel solution for creating turbulence in wind tunnels. Provide one or more references to previous work.

**Response:** We essentially want to produce a flow that leads to a different distribution of surface shear stress. In the experiment, we measured the distribution of surface shear stress and verified the effect of this cloth. We speculated that the effect of cloth is not only to create turbulence but also to cause a flow effect similar to convection. We didn't focus on the details of flow because they are not the main topic of this article. However, thanks for the reviewer's reminder, we have adjusted the expression of this part to make it more appropriate. Previous works of convection simulation (**water tanks**, e.g., Willis and Deardorff, 1974, Yuan et al., 2013; **saline tanks**, e.g., Hibberd and Sawford, 1994; **thermally-stratified wind tunnels**, Hancock and Hayden, 2018) in the laboratory have been added as references. In relation to comments iv), it is important to note that for conventional wind tunnels, to the best of our knowledge, there have been no previous reports of simple ways to generate turbulence similar to convective turbulence in terms of probability density function has positive skewness. The positive skewness of convective turbulence plays an important role in aeolian particle entrainment. We added the above works

in the introduction of the revision (highlight version) in line 55-63:

"Wind tunnel is a powerful tool for studying aeolian problems under controlled flow conditions (e.g., Rasmussen and Mikkelsen, 1991; Alfaro et al., 1997; Brown et al., 2008; Zhang et al., 2014). Although several methods have been proposed to generated turbulence in wind tunnels, including spires, roughness element, grid etc., all these methods are designed to increase he intensity of turbulence in neutral ABLs, but are inadequate for generating large eddies similar to those commonly seen in convective ABLs. To generate convective turbulence inn wind tunnels usually requires the use of additional thermal forcing from the surface (e.g., EnFlo stratified flow wind tunnel, Hancock et al., 2013; Hancock and Farr, 2014; Hancock and Zhang, 2015; Hancock and Hayden, 2018), temperature control of recirculating air and floor panels (e.g., Inagaki et al., 2012; Zhang et al., 2013; Kanda and Yamao, 2016), thermally stratified wind tunnels (e.g., Marucci et al., 2018; Marucci and Carpentieri, 2020), etc. To apply surface heating requires normally a very large wind tunnel."

ii) Details of the flapping cloth are missing. What type and weight of fabric? Width is given but more importantly, how long was the sheet? What was the wavelength, amplitude and frequency of its oscillation?

**Response:** Thanks for the suggestion. We have added the details of the cloth, including type, weight, and size in the revision (highlight version) in line 78:

"The cloth is a woven fabric (grammage 200 g m$^{-2}$) with size of 1 m in width and 1.5 m in length."

Since how the cloth changed the flow was not the focus of this experiment, the wavelength, amplitude, and frequency of the oscillation were not investigated and recorded in the experiment. But this is a very interesting topic, we can do some special research in the future work.

iii) Rationale is not provided for the placement of the cloth relative to the test surface. Is the elevation scaled with the length of the cloth or distance from the leading edge of the sample tray?

**Response:** As mentioned above, the focus of this paper is not to study the effect of cloth on flow. What we're more interested in at that time is whether this cloth in the wind tunnel can provide the conditions that we need for our experiment, i.e., whether it can produce quasi-convective flow and shear with different distributions on the surface. We did a series of tests before the experiment to check the effect of the cloth, and empirically determine the length of the cloth (1.5m), the height of the arrangement (0.7m), and the distance from the leading edge of the sample tray (1.5m). We have added it in the revision (highlight version) in line 79-80:

"The cloth size was empirically determined by a series of tests before the formal experiment, to satisfy the requirement on generating quasi-convective turbulence."

Of cause, the rationale for the placement of the cloth relative to the test surface insignificant and deserves further study.

iv) Spires are much more commonly used to generate large scale eddies in wind tunnel simulation work and these are typically placed upwind of the roughness element array. There is also a large literature on the effect of spires on turbulent flow and shear. Why did you elect to use a flapping cloth over spires?

**Response:** It is indeed several methods have been proposed to generate turbulence in wind

tunnels, including spires, roughness elements, grid, etc., as we answered in i). But all of these methods are for neutral boundary layers, and inefficient for the simulation of large eddies commonly observed in convective ABLs. To generate convective turbulence usually requires the use of additional thermal forcing from the surface (**EnFlo stratified flow wind tunnel**, Hancock et al., 2013; Hancock and Farr, 2014; Hancock and Zhang, 2015; Hancock and Hayden, 2018; **controlling temperature of recirculating air and floor panels,** Inagaki et al., 2012; Zhang et al., 2013; Kanda and Yamao, 2016; **thermally-stratified wind tunnels**, Marucci and Hayden, 2018; Marucci and Carpentieri, 2020). To apply surface heating requires normally a large wind tunnel. In any case, to the best of our knowledge, studying the effect of convective eddies on aeolian processes has never been done in wind-tunnel experiments, mainly because we have so far no adequate means to generate convective turbulence in a wind tunnel for aeolian experiments.

For this work, our hypothesis is that the change in the distribution model of surface shear caused by large eddies (like convection) may significantly influence aerodynamic entrainment, and this influence could not be determined by the mean surface shear. Therefore, we need to first simulate flow conditions similar to convection to produce a distinct distribution of surface shear in the wind tunnel, which is the reason for the use of the cloth. Although we have not studied the intrinsic mechanism of how cloth induces large eddies, so far, it does not affect our research on the quantitative characterization of aerodynamic entrainment. This study does provide an inexpensive and practical way to generate the quasi-convective eddies without involving surface heating. Some discussion is added in the revision (highlight version) in line 63-64:

"To the best of our knowledge, there has been no previous report of simple ways to generate turbulence similar to convective turbulence which in terms of probability density function has a positive skewness which plays an important role in aeolian particle entrainment."

**Comment 16:**

**Pitot tubes:**

i) Provide dimensions. Pitot tubes come in a range of sizes, while large tubes places in a fixed vertical array can initiate flow stagnation – bluff body effect.

ii) What did you sample the pressure difference with? State precision and sampling time.

**Response:** Thanks for your suggestion. There are two types of pitot tubes used in this work. The outer diameter of the small pitot tubes is 1 mm, and the inner diameter is 0.5 mm. Nine these pitot tubes make up a wind profiler, which could simultaneously measure the wind speed profiles. A bigger one with an outer diameter of 5 mm and an inner diameter of 2 mm is used to standardize all the small pitot tubes. We have added it in the revision (highlight version) in line 85-93:

"The anemometers, including the hotwire (1D, fixed at 10 mm height and employed only in clear air condition) and the wind profiler (combined by nine pitot tubes placed at the levels of 6.5, 10, 15, 30, 60, 120, 201, 351 and 501 mm), were located between the trays. The outer diameter of the pitot tubes in the wind profiler is 1 mm, and the inner diameter is 0.5 mm. The wind profiler measures the profile of the mean flow speed with a sampling frequency of 1 Hz, while the hotwire anemometer measures turbulent fluctuations with a sampling frequency of 1000 Hz."

**Comment 17:**

**Measurement of the mass transport rate:**

i) One of the greatest challenges with accuracy in measuring mass in the lab using an electronic balance is drift associated with changing air pressure in the room. So it is possible that any instantaneous fluctuation in F is the a consequence of both the change in mass of particles in the tray and the pressure perturbation in the flow. This would be particularly exacerbated by the presence of large scale eddies. Please explain in detail how you accounted for this in your analysis. Was the total mass loss obtained from measurements in the absence of an airflow or did you average the instantaneous mass transport rate throughout the test?

**Response:** Sorry for we made this misunderstanding. In fact, we did not weight the tray instantaneously. The tray is only weighted after a test to calculate the average entrainment rate. We modified it in the revision (highlight version) in line 83-84: "Each tray will be weighted before and after each test by an electronic balancer with a precision of 0.01 g in the range 5 kg, to determine the net mass loss of the tested surface."

Hot wire anemometer and Irwin sensors:

ii) Please state in line 70 that the instrument only samples in 1D

Response: Thanks for pointing this out, we have stated it in the revision (highlight version) in line 85:

"The anemometers, including the hotwire (1D, fixed at 10 mm height and employed only in clear air condition) and…"

iii) Was the wire ruggedized to withstand particle impact during saltation? If so, how did this affect sensitivity (time constant and precision)?

**Response:** The wire anemometer was used only in the clean flow to protect from particle impact.

iv) Please provide exact dimensions of the Irwin sensor rather than a citation. The height of the central port does vary from study to study and is important to know relative to the roughness of the bed surface. Describe also the roughness of the bed surface upwind of the sample trays. Was it smooth or roughened to give a suitable aerodynamic roughness matched to that of the sand in the tray?

**Response:** The diameter of the inner port is 1.65mm, and 1.75mm in height; the diameter of the outer port is 2.57mm; the diameter of the sensor is 12.5mm. The bed surface upwind is covered by sandpaper (40 mesh). We added all these information in revised Fig. 1:

[Figure]

**Figure 1:** (a) Top view of the wind-tunnel configuration; (b) Side view of the wind-tunnel configuration, a piece of randomly fluttering cloth in the wind tunnel enables the generation of quasi-convective turbulence. (c) The appearances of employed probes. For the Irwin sensor with diameter of 12.5 mm, the diameter of inner port is 1.65 mm, and 1.75 mm in height, the diameter of the outer port is 2.57 mm. (d)The size distributions of 4 tested soils.

and text in line 75:

"The rest section is covered by a 40-grit sand paper to simulate a non-erodible sandy surface."

**Comment 18:**

**Table 1**

i) Why so many missing experiments? Indeed, there are so few experiments at the 0.5 X1000 rpm increment that it might be better to exclude these data altogether. Add the freestream velocity values for the NP experiments, as rpm will make little sense to the reader.

**Response:** Actually, the cases of 0.5 X1000 rpm are supplementary tests. In the case of large surface shear, the erodible surface (S2 and S3) rapidly appeared obvious surface-concave which would affect the test results. So, we added several tests in low surface shear. We have explained this in the revision in the note of Table 1:

"The tests of 0.5×1000 rpm are supplementary. In the case of large surface shear, the erodible surface (S2 and S3) rapidly appeared obvious surface-concave which could affect the test results. We therefore added several tests for low surface shear."

We have added the axis wind speed of the incoming flow in Table 2.

ii) Describe the bed surface composition for your control runs without a tray/soil.

**Response:** Sorry for the loss of relevant information, we added the description of the tunnel floor in revised Fig. 1 and text in line 75:

"The rest section is covered by a 40-grit sand paper to simulate a non-erodible sandy surface."

**Comment 19:**

**Figure 3**

i) I am not clear on the surface/sediment texture characteristics pertaining to these data. Have the data for all textures been lumped together?

**Response:** Figure 3 illustrates the average wind speed profiles over the wind tunnel floor for all the tested cases. There is only one texture of the wind tunnel floor and the tested particles are filled in trays mounted flush to the tunnel floor. The tray is small in area and the surface is smoothed, so we believe that its influence on the wind profile is very limited. We have added these descriptions of experimental conditions to make this information clearer in the revision (highlight version) in line 75-85:

"The rest section is covered by a 40-grit sand paper to simulate a non-erodible sandy surface.…Two sand trays [285 mm wide, 150 mm long and 13 mm deep, which have been tested as a suitable option for the study of aerodynamic entrainment (Li et al., 2020)] are placed 1.5 m downstream the end of the fluttering cloth. The trays filled with sand are mounted flush to the tunnel floor. The sand surface is smoothed before every test. Each tray is weighted before and after each test by an electronic balancer with a precision of 0.01 g in the range 5 kg, to determine the net mass loss of the tested surface"

ii) The NP vertical profiles of the total velocity look great – a deep well-structured boundary-layer flow following the law of the wall. The WP profiles also look very good, but I think even

more might be inferred from them than what is provided. There is indeed a high degree of flow stagnation in the upper part of the profile above 20 cm associated with partitioning of momentum to the flapping cloth turbulent but also turbulent energy dissipation within the wake flow downwind. However, little is said about the proportionate acceleration of the air flow (~0.5 m/s) at all levels below 20 cm in all WP experiments. This is unexpected and could be a wind tunnel artifact associated with a small degree of compression of the flow that was redirected beneath the cloth and between the confining walls.

**Response:** Yes, for the cases of NP, the wind profiles are normal. According to the analysis of regression based on the logarithmic law (Eq. 5 with $\Psi_m = 0$), we obtained the friction velocity for each case to calibrate the Irwin sensor.

When we use the same method to analyze the data of WP conditions, there is a great difference between the estimated friction wind speed and the ones measured by the Irwin probe. However, if we take the effect of $\psi_m$ into account in the regression equation (i.e., not set to zero), the obtained friction is very close to Irwin's result. Therefore, we speculate that the cloth induces a flow similar to the convection in ABL.

We noticed an increase in wind speed near the surface, which we think is a result of the presence of the cloth increasing the exchange of horizontal momentum in the vertical direction.

Unfortunately, we have not measured more wind field information besides the wind speed profile (Pitot tube) and the time series of the wind speed at a certain height (1-dimensional hot wire), so it is impossible to do a deeper analysis of the turbulence structure induced by the cloth.

iii) It is really unfortunate in this study that the vertical component of the total wind speed was not isolated and sampled, as required for analysis of the eddy structure. As first identified by Thom (1975) for truly unstable conditions with convection, mechanical effects (roughness) dominate the near surface flow, but in moving away from the surface the eddies are stretched vertically and the momentum flux is enhanced. Since the vertical velocity is generally one to two orders of magnitude smaller than the horizontal component, the total wind speed as sampled in this study cannot be particularly sensitive to the eddy perturbation that was intentionally created.

**Response:** Thanks for your suggestion. The simulation and analysis of convective flow in the wind tunnel is indeed an interesting and challenging topic. We will continue to focus on this in the future.

**Comment 20:**
**Table 2**
i) Again, I am not clear on whether the data have been averaged across all sediment textures or this is just one example for a specific texture.

**Response:** As we responded above, there is only a smoothed and mounted sediment surface with a small area employed to test. The wind field is mainly affected by the wind tunnel floor which is unchanged during the experiment.

ii) Why is the threshold friction velocity also not reported here?

**Response:** Table 2 only provides the parameters of the wind field. The threshold friction velocity is a property parameter of the grain surface, which is provided in Table 3.

iii) Over what elevations were the wind speed data used to calculate u*? If the boundary layer

depth is taken to be 20 cm for the WP experiments and the inner constant stress region 15-20% of that, then your calculation might only be based on 4 data points. That is, the shear stress values for the WP experiments would carry greater error than those for the NP experiments. How does this uncertainty compare with the roughly 7% increase in u* associated with introduction of the flapping cloth?

**Response:** Thanks for your suggestion. There is indeed uncertainty in estimating friction velocity from wind profile. In fact, the shear stresses or friction velocities in Figs. 4-7 are provided by Irwin probe data.

iv) Under what conditions were the Irwin sensors calibrated – NP or WP? This has a large bearing on an explanation for why the Irwin sensor u* values sit slightly higher than for the WP data.

**Response:** Irwin sensor is calibrated under NP conditions which have been validated for well estimating the friction velocity from the wind profile.

**Comment 21:**

Line 171

"Convective turbulence is much more efficient in lifting particles into the air". Caution should be exercised in making such a broad statement. This may well be true for very light particles (aerosols) entering suspension, given a settling velocity that is lower than the vertical velocity at which eddies burst away from the bed surface, but not so for very large sand particles. Once again, it is unfortunate that you can only infer such effects because you don't have the vertical component measurements.

**Response:** Thanks for the comment. Our results on four tested surfaces with different average particle sizes show that convection facilitates particle entrainment. We prove that this is not only caused by the increase of mean surface shear stress, but also by the change in the distribution of surface shear stress. Anyway, we added more discussion about it in the revision (highlight version) in line 217-220:

"It shows that the slight increase of $\bar{\tau}$ in quasi-convective conditions is not sufficient alone to explain the measured differences in the entrainment rates of the four soils. It implies that the perturbations of the shear stress, $\tau'$, are also responsible for a part of the differences in the entrainment rates."

To make it clearer.

**Comment 22:**

Figures 4 and 5

There is no question that positive skewness in the instantaneous bed shear stress should increase the particle entrainment rate at low values near threshold, but the fact that the near surface flow is accelerated (especially at lower elevations) by insertion of the flapping cloth is likely to be equally if not more important in this particular study.

**Response:** Thanks for the comment. We believe that the increase of wind speed in the near-surface region is reflected in the increase of surface average shear stress. But the increase in average shear cannot completely correspond to the enhanced particle entrainment (Fig. 4). Only by taking the influence of the change of shear stress distribution into account, the measured results of entrainment are well explained.

**Comment 23:**

**Conclusions**

Logic dictates and your study shows that intermittent generation of large shear stresses on the bed surface enhances the entrainment of sand and silt sized particles. This should be qualified by some form of statement to indicate that this effect is greatest at low wind speeds around threshold and when transport is intermittent. In really strong winds mechanical or impact entrainment dominates, while the particle borne stress largely determines the transport under saturated conditions.

**Response:** Thanks for the comment. Fig. 7 obviously shows that the effect of the distribution of surface shear is greatest at low wind speeds around the threshold and when transport is intermittent. We have added more statement in the revision (highlight version) in line 258-261:

"We can thus conclude that convective turbulence may significantly enhance dust entrainment by alter how shear stress acts on the surface, especially for the cases of intermittent entrainment when the mean shear stress is below the threshold. Considering that the lower-than-threshold wind conditions may have a high temporal weight in natural conditions, we believe that this effect deserves particular attention in dust emission schemes."

And line 290-292:

"This enhancing effect is greatest at low wind speeds around threshold and when transport is intermittent, and becomes weaker when wind speed is strong enough, while the particle borne stress largely determines the transport under saturated conditions."

"we showed that wind-tunnel turbulence lacks energy containing eddies even compared with ABL flows in neutral conditions, highlighting the deficiency of traditional wind-tunnel experiments on aeolian studies". Again, this is a bit of an overstatement. Most traditional wind tunnel studies have modelled saturated flows (see above) where the particle cloud itself is so dense that it alters the turbulence structure. Since the development of fast response pressure transmitters, cross-wire probes, and laser Doppler anemometers more than 2 decades ago, aeolian researchers working in wind tunnels have indeed provided detailed measurements of the turbulence structure and intensity associated with sediment entrainment and transport while also investigating the effects of wind gusting (e.g. Li and McKenna Neuman 2012, 2014 etc). Similarly, there are many means by which we can and do generate large scale eddies in wind tunnel simulation and these are widely practiced in engineering applications, particularly in investigations of wind loading on urban structures.

**Response:** Thanks a lot for the comment and for providing information. We have modified the relevant part to make it more rigorous in the revision (highlight version) in line 275-280:

"Although advanced technology for measuring turbulence structure and intensity associated with sediment entrainment and transport are developed in the past few decades, some focusing on the effects of wind gusting (e.g., Li and McKenna Neuman, 2012; Li and Neuman, 2014), few studies have been done focusing on the effects of the ABL convective turbulence. Also, means for generating large scale eddies (not necessarily convective large eddies) in wind tunnel simulation for wind engineering applications, haven't been widely used in aeolian studies due to their complexity and high operational cost."